# Broad phylogenetic analysis of cation/proton antiporters reveals transport determinants

Gal Masrati[1], Manish Dwivedi[2], Abraham Rimon[2], Yael Gluck-Margolin[2], Amit Kessel[1], Haim Ashkenazy[3], Itay Mayrose[4], Etana Padan[2] & Nir Ben-Tal [1]

Cation/proton antiporters (CPAs) play a major role in maintaining living cells' homeostasis. CPAs are commonly divided into two main groups, CPA1 and CPA2, and are further characterized by two main phenotypes: ion selectivity and electrogenicity. However, tracing the evolutionary relationships of these transporters is challenging because of the high diversity within CPAs. Here, we conduct comprehensive evolutionary analysis of 6537 representative CPAs, describing the full complexity of their phylogeny, and revealing a sequence motif that appears to determine central phenotypic characteristics. In contrast to previous suggestions, we show that the CPA1/CPA2 division only partially correlates with electrogenicity. Our analysis further indicates two acidic residues in the binding site that carry the protons in electrogenic CPAs, and a polar residue in the unwound transmembrane helix 4 that determines ion selectivity. A rationally designed triple mutant successfully converted the electrogenic CPA, EcNhaA, to be electroneutral.

[1] Department of Biochemistry and Molecular Biology, George S. Wise Faculty of Life Sciences, Tel-Aviv University, Ramat-Aviv, 69978 Tel-Aviv, Israel.
[2] Department of Biological Chemistry, The Alexander Silberman Inst. of Life Sciences, The Hebrew University of Jerusalem, Jerusalem 91904, Israel.
[3] Department of Cell Research and Immunology, George S. Wise Faculty of Life Sciences, Tel-Aviv University, Ramat-Aviv, 69978 Tel-Aviv, Israel.
[4] Department of Molecular Biology and Ecology of Plant, George S. Wise Faculty of Life Sciences, Tel-Aviv University, Ramat-Aviv, 69978 Tel-Aviv, Israel.

Many biological processes depend on pH homeostasis and ion concentrations. It is thus not surprising that cation/proton antiporters (CPAs) are prevalent in almost all living species. These antiporters mediate the exchange of monovalent cations, mainly $Na^+$ and $K^+$, with one or two protons across the membrane. In humans, 13 different CPAs have been identified and deficiencies in CPAs are related to pathologies, ranging from hypertension[1,2] to autism spectrum disorders[3] and cancer[4].

CPAs can differ significantly in their sequence, but the available high-resolution structures[5–8] show that they share a similar transmembrane topology, known as the NhaA fold[9]. The fold is organized in two functional domains—a dimerization domain and a conserved core domain encapsulating the ion binding site[10]. The core domain includes two unwound transmembrane helices (TMs) that cross each other in the middle of the membrane near the ion binding site, creating an x-shaped structure, characteristic of the NhaA fold (Supplementary Fig. 1)[10].

Functionally, CPAs are often classified based on two phenotypes, their ion selectivity, i.e., whether they transport $Na^+$ or $K^+$, and their electrogenicity, i.e., whether a cation exchanges for one (electroneutral) or two (electrogenic) protons. Based on experiments with *Escherichia coli* NhaA (EcNhaA), it has been suggested that electrogenic CPAs are characterized by two conserved aspartates in their ion binding site—D163 and D164 on TM-5 in EcNhaA. D164 has been proposed to be the primary proton carrier in both electroneutral and electrogenic transporters, while D163, absent in electroneutral CPAs, is thought to be the second proton donor in electrogenic antiporters[11–13]. However, an alternative antiport mechanism has been recently proposed[14,15], pointing to a conserved lysine close to the ion binding site as the second proton carrier (K300 on TM-10 in EcNhaA). In its protonated form, the lysine salt bridges with D163. Upon breakage of the bridge following ion binding, it supposedly deprotonates, releasing the second proton. This model also suggests that in electroneutral transporters, the lysine is replaced with arginine, which remains protonated due to its high pKa and cannot facilitate the transport of a second proton. Only a few studies have investigated the molecular basis for the ion selectivity of CPAs. Site directed mutagenesis of the sodium-selective SOD22 from *Zygosaccharomyces rouxii* (ZrSOD22) suggested that a hydrophobic filter near the transporter binding site confers selectivity. This putative filter includes residues from the two unwound helices (TM-4 and TM-11)[16,17].

Previous phylogenetic studies have mainly surveyed the evolutionary relationships between CPAs of limited diversity[18–21]. These analyses divided CPAs into CPA1 and CPA2, which occasionally are stated to relate to the phenotypical electroneutral/electrogenic partition[6,8,22,23]. Specifically, the common idea is that CPA1s are electroneutral, while CPA2s are electrogenic. However, this partition has yet to be truly established, and there is still much debate concerning the mechanism behind electrogenicity, as the data are conflicting[13–15,24]. Additionally, the exact molecular determinants that confer ion selectivity are still unclear.

Exploiting the recent flood of protein sequence, we study the evolutionary relationships among 6597 transporters that encompass the enormous richness of CPAs. We reveal a well-defined sequence motif that distinguishes CPA1s from CPA2s and appears to determine the characteristics of electrogenicity and ion selectivity. Our findings imply that the phylogenetic division of the CPA superfamily only partially corresponds with the functional electroneutral/electrogenic partition, in contrast to previous suggestions. Finally, to experimentally test our computational analysis, we design a triple mutant that rescues an inactive EcNhaA variant, further supporting the importance of two acidic residues in the binding site to electrogenicity.

## Results

**Reconstructing the phylogenetic tree**. We aligned a seed group of 146 CPAs that share the NhaA-fold and produced a profile hidden Markov model (HMM) of their membrane segment using HMMER-3.1[25]. The profile HMM was then used in a HMMER search to broaden the sequence pool. Highly similar sequences were removed, and each of the remaining sequences was aligned to the HMM profile. The enriched multiple sequence alignment (MSA) was subsequently used for two independent phylogenetic analyses. A fast approximation of the unrooted phylogenetic tree was performed using FastTree 2[26], and a more rigorous tree was reconstructed using IQ-TREE-1.6.2[27]. The two different methods divided the CPA superfamily into the same main clades (Supplementary Fig. 2), and the results presented here are based on the IQ-TREE analysis. Bootstrap analysis was conducted with 100 replicas. To improve phylogenetic robustness, dropsets of up to two sequences that assumed different positions in different trees and reduced the overall bootstrap support were removed using RogueNaRok[28], while larger dropsets were removed manually. The final number of proteins amounted to 6537. From these 6537 representatives, we selected the 500 most divergent sequences using the phylogenetic diversity analysis tool PDA-1.0.3[29]. Then, an additional IQ-TREE analysis was performed to further asses the robustness of our results.

**Phylogenetic tree**. The reconstructed phylogeny divided the CPA sequence pool into two groups, with high bootstrap values of 93%. By mapping proteins that are classified as CPA1 and CPA2 onto the tree, we conclude that the observed two groups reflect these two main CPA subtrees (Supplementary Fig. 3). Moreover, some distinct clades within the tree appear to consist of proteins that share electrogenic properties and/or are potassium selective. Thus, the phylogenic tree appears to reflect the main characteristics of the CPA superfamily.

The CPA1 subtree can be further divided into six main clades that appear to be electroneutral, all supported by high bootstrap values (Fig. 1). The first of these was designated the NhaP-I/NHE clade, as it includes bacterial $Na^+/H^+$ antiporters (NhaP) alongside eukaryotic CPAs, such as mammalian NHEs. The second, named the NHA clade, includes $Na^+/H^+$ antiporters from prokaryotes alongside eukaryotic genes from fungi. This clade was previously assigned to the CPA2 sub-tree[21]. However, based on the conserved sequential features discussed below, the assignment of fungal NHAs to the CPA1 sub-tree is more appropriate. The third, NhaP-II $K^+$-specific clade, features orthologues of CPAs that are specific for potassium ions[30,31]. Members of the fourth, named the NhaP-III clade do not include any well-studied transporters and their exact characteristics are unknown. Finally, the fifth and sixth groups were designated archaeal- and bacterial-NhaP-II $Na^+$-specific clades. The former includes the archaeal $Na^+(Li^+)/H^+$ antiporters *Methanocaldococcus jannaschii* NhaP1 (MjNhaP1) and *Pyrococcus abyssi* NhaP (PaNhaP), the structures of which are known[7,8]. An additional clade of 17 sequences could not be characterized.

The CPA2 subtree comprises nine main clades (Fig. 2). The first, named the NhaA clade (bootstrap 99%) includes the well-studied EcNhaA, which mediates electrogenic transport with a $1Na^+:2H^+$ stoichiometry[32]. The high sequence similarity among the orthologues in this clade suggests that they all are electrogenic. The second and third clades are NapA-I and NapA-II (bootstrap 94% and 80%, respectively). *Thermus thermophiles* NapA (TtNapA), also electrogenic[6], is part of NapA-I. The high sequence similarities between NapA-I/II genes and the members of a fourth CPA2 clade, named the GerN clade (bootstrap 73%) suggest that these groups may all be electrogenic[33]. However, focusing on the 500 most

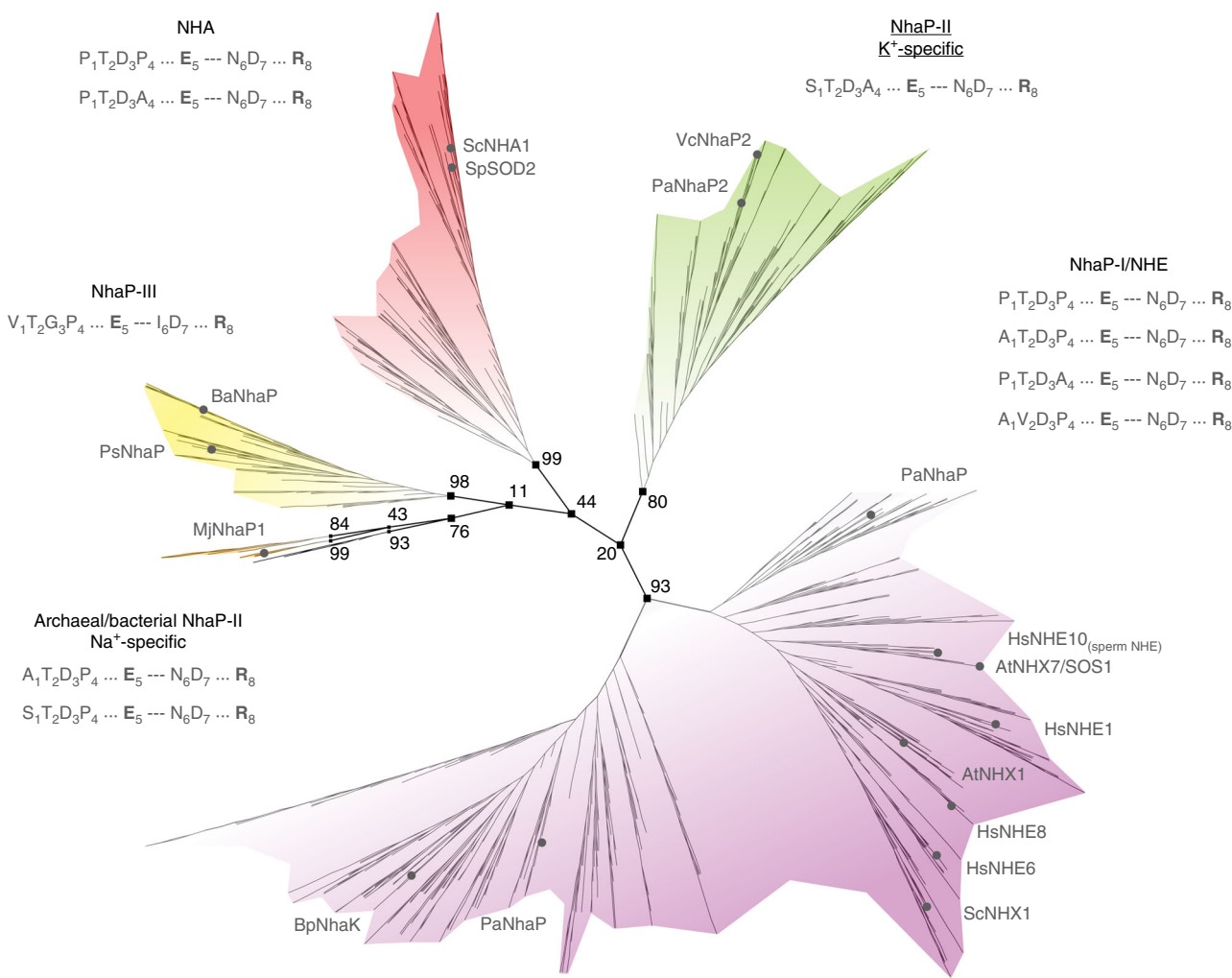

**Fig. 1** The CPA1 subtree. Unrooted tree with the main clades highlighted in different colors. The names of $K^+$-selective clades are underlined. The prevalent motifs that correlate with each clade are listed. The invariable positions 5 and 8, a fingerprint of all CPA1s, are highlighted in bold. Bootstrap values are marked for the branches that separate the main clades. Names of representative CPAs are presented at the leaves

diverged sequences out of all analyzed sequences resulted in low bootstrap value for the GerN clade (10%). A fifth CPA2 clade includes plant CHX-like genes (bootstrap 80%). Whether plant CHXs are electrogenic or not has yet to be established. The sixth includes potassium-selective efflux (Kef) transporters, and was designated the Kef-like clade (bootstrap 90%). Experiments with a purified variant of this group, reconstituted into proteoliposomes, suggest that at least some mediate electroneutral transport[34]. The seventh, designated the KhaB clade (bootstrap 84%), includes prokaryotic CPAs, such as *Synechocystis sp.* NhaS5 (SsNhaS5). Analysis of mutants that lack SsNhaS suggests that they may exchange potassium ions[35]. The eighth, designated the animal-NHA-like clade (bootstrap 99%), includes prokaryotic and eukaryotic CPAs, including the only animal CPA2s. A member of the latter, human NHA2 (HsNHA2), mediates electroneutral transport[24]. The ninth clade (bootstrap 69%) includes prokaryotic CPA2s with no significant similarity to any well-characterized transporter. More detailed description of the tree is provided in Supplementary notes.

**Eight specificity-determining amino acids.** Because the tree can reproduce the CPA1/CPA2 division, with the exception of fungal NHAs, and can discriminate between-specific clades with distinct phenotypes, we looked for a minimal set of amino acids that form

the basis of these different phenotypes. We used ConSurf to identify structurally- and functionally important residues which tend to be evolutionarily conserved among homologous proteins[36]. In order to focus on the function and the transport mechanism, we looked for charged or polar residues, which are generally less prevalent in membrane proteins[37]. The high-energy penalty associated with the placement of such residues in the lipid membrane suggests their importance. We then annotated each sequence with the residues populating these conserved positions and mapped this information onto the tree (Figs. 1, 2) searching for associations between sequential features and clades with distinct functional characteristics.

Despite the high diversity among CPAs, we were able to identify a minimal set of eight highly conserved positions, which seems to capture the different CPA clades (Figs. 1, 2) and may lay the basis for electrogenicity and ion selectivity. Notably, all eight positions clustered at the protein core and included the substrate binding site (Fig. 3a, b). However, they reside on different helices and do not form a continuous sequential motif (Fig. 3c). The motif can be represented as $X_1X_2X_3X_4 \ldots [E/-]_5 - - - X_6D_7 \ldots [R/K]_8$, where X stands for a limited set of residues that characterize each group, the dashes represent any residue, and the dots represent two segments with average lengths of 23 and 156 residues, respectively. The subscripts are one through eight index

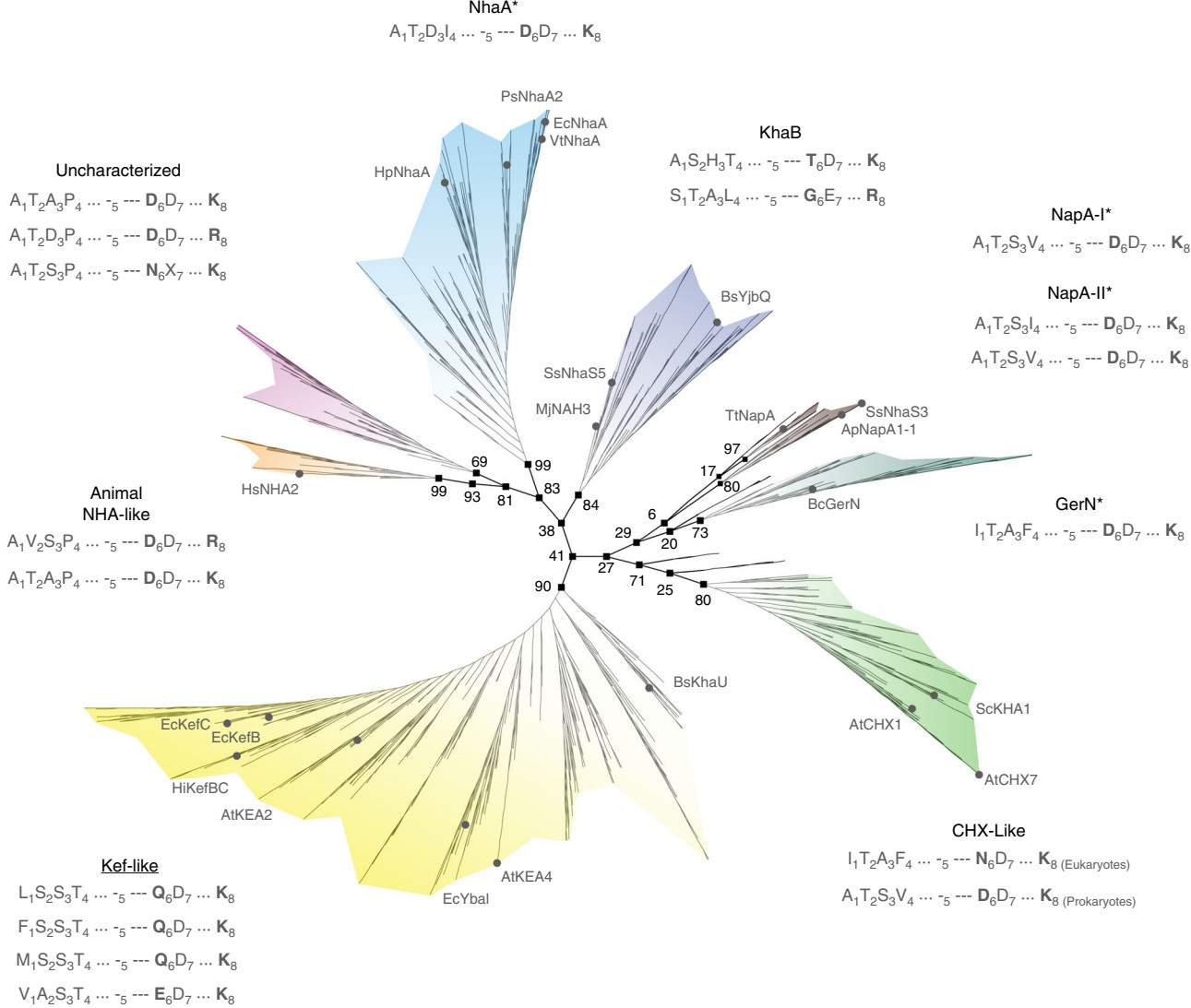

**Fig. 2** The CPA2 subtree. Unrooted tree with the main clades highlighted in different colors and the prevalent motifs that correlate with each clade listed as in Fig. 1. The names of K$^+$-selective clades are underlined; whereas. the names of known electrogenic clades and sub-clades are marked with an asterisk. Bootstrap values are marked for the branches that separate the main clades. Names of representative CPAs are presented at the leaves

numbers, corresponding to the motif residues. Below we refer to the relevant index number alongside the helices and amino acids numbering in EcNhaA, and to the motif as the CPA motif.

Positions one-through-four of the CPA motif correspond to the unwound section of TM-4, EcNhaA A131, T132, D133, and I134 (Fig. 3a, c). They feature a limited variety of residues that distinguish between different CPA groups within each of the two main clades. The fifth position, marked as $[E/-]_5$, is located on TM-5 (Fig. 3). It is highly conserved only among CPA1s, where it accommodates only glutamate (Fig. 3b, see also ref. [38]), while in CPA2s it can feature any residue. Therefore, we refer to it as $E_5$ when discussing CPA1s and $-_5$ in the context of CPA2s. The sixth and seventh positions, D163 and D164, also on TM-5, comprise the well-known ND/DD motif[12] in the binding site (Fig. 3). D164 is conserved in 94% of the sequences and has been repeatedly shown to be essential for transport[39–41]. The eighth position features mostly arginine or lysine (95% of the sequences) and is located on TM-10 (K300; Fig. 3). Consistent with our findings, these eight positions were recently proposed to be important for the function and regulation of eukaryotic and archaeal CPA1s, based on a few representatives[38].

**The CPA1/CPA2 division**. Projecting the CPA motif onto the superfamily's phylogeny revealed four sequence variations involving position 5 and 8, or 6 and 8 of the motif that are conserved among 97% of the sequences (Figs. 1, 2). Based on existing structures, these sequential features may reflect stabilizing electrostatic interactions between TM-5 and TM-10 in the core domain (Fig. 3a, b). One of these sequence variations exists exclusively in CPA1s, discriminating them from CPA2s. In our analysis, virtually all CPA1s feature a glutamate at position 5 of the CPA motif (TM-5), one helix-turn below the binding site, and a conserved arginine at position 8 (TM-10; Figs. 1, 3b). In the two available structures of CPA1 members[7,8] these positions are salt bridged (Fig. 3b). This feature is also found in fungal NHAs, which were previously classified as CPA2s[21]. Based on our work, these transporters should be assigned to the CPA1 sub-tree. The ND motif at positions 6 and 7, often considered a marker for CPA1s[38], is indeed conserved, but not exclusively in CPA1s, as it also appears in some CPA2s (Fig. 2).

CPA2s lack the conserved glutamate at position 5 (TM-5) that characterizes CPA1s. However, the putative electrostatic interaction between TM-5 and TM-10 could potentially include position

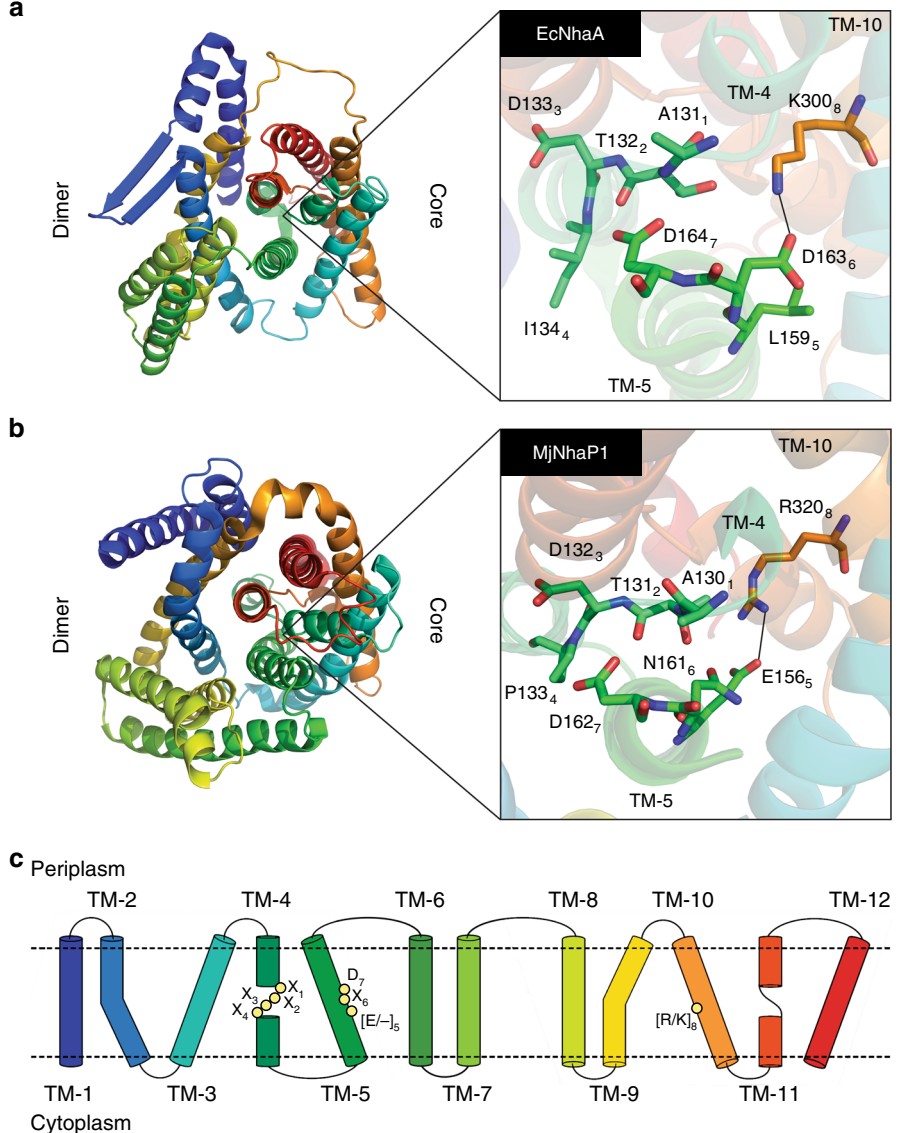

**Fig. 3** The CPA motif is located in the core domain, near the ion binding site. **a** Periplasmic view of the electrogenic EcNhaA, showing the dimerization domain to the left and core domain to the right. The frame to the right displays a close-up view of the motif. The eight residues of the motif are shown using sticks representation, each with its sequence number and corresponding position in the motif (subscript). Their respective helices are designated in italics, and the conserved polar interaction between TM-5 and TM-10 is marked with a solid line. **b** The electroneutral MjNhaP1, shown as in **a**. The numbered residues correspond to their equivalents in EcNhaA. For example, D132 and E156 in MjNhaP1 correspond to D133 and L159 in EcNhaA, respectively. **c** Schematic two-dimensional representation of EcNhaA's structure. The membrane boundaries are shown as dashed lines, the helices are numbered TM-1-through-TM-12, and the motif residues are indicated with yellow spheres

6 and 8 of the CPA motif, and if so, the nature of it varies within the clade. In EcNhaA (NhaA clade) and TtNapA (NapA-I clade), which are both electrogenic, position 8 is populated by lysine (K300), shown to salt bridge with aspartate at position 6 (D163) (Figs. 2, 3a)[6,14,15]. In other CPA2s, e.g., some of the CHX and Kef-type transporters, this aspartate is replaced with asparagine or glutamine (Fig. 2). In such cases, hydrogen bond(s) could substitute the salt bridge, though there is no structural data to support this yet. Overall, CPA2s are more diverse in the CPA motif compared to CPA1s (Figs. 2 vs. Fig. 1).

**Electrogenicity.** Based on the phylogeny and available experimental data, we could identify a minimum of four electrogenic clades, all of which are CPA2s: The NhaA, NapA-I, NapA-II, and GreN clades[6,32,33] (Fig. 2). The phylogeny further suggests that

these electrogenic transporters are characterized by an aspartate at position 6 of the CPA motif and a lysine at position 8 (Fig. 2). However, these features are not shared by all CPA2s (Fig. 2).

**Ion selectivity.** The tree features two main $K^+$-selective clades: CPA1 NhaP-II-$K^+$-specific, and CPA2 Kef-like antiporters. In comparison to $Na^+$-selective clades, the difference can be traced back to positions 1-through-4 in the CPA motif comprising the unwound section of TM-4 in EcNhaA (Fig. 3a, c). Position 1 in CPA1 $Na^+$-selective clades, such as the NhaP-I/NHE clade, and position 4 in CPA2 $Na^+$-selective clades, like the NhaA clade, are populated by non-polar residues. For example, in NhaP-I/NHE antiporters, positions 1-through-4 form a $P_1T_2D_3P_4$ motif (Fig.1). Likewise, the $Na^+$-selective CPA2s of the NhaA clade share an $A_1T_2D_3I_4$ motif (Fig. 2). On the other hand, in $K^+$-selective

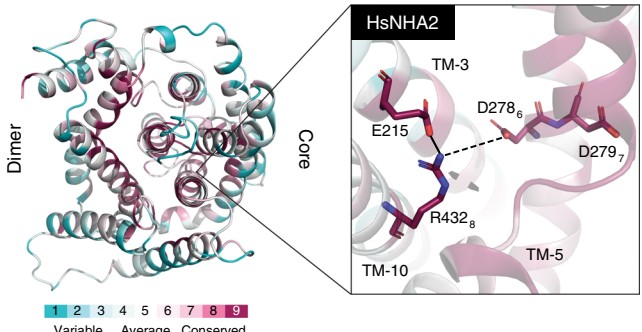

**Fig. 4** Model structure of HsNHA2. Left: periplasmic view, colored by the ConSurf evolutionary conservation bar at the bottom. The dimerization domain is on the left and core domain on the right. Right: close-up view of the unique polar interaction between TM-3, TM-5, and TM-10 characteristic of HsNHA2-like PCAs. The three residues involved are highly conserved. E215 could potentially salt bridge with $R432_8$ on TM-10. In one plausible scenario $D278_6$ would be protonated and hydrogen bonded to $R432_8$, leaving only $D279_7$ to alternate its protonation state upon transport, resulting in electroneutral transport. Modeling is based on the structure of TtNapA (PDB id 5BZ2) as template

antiporters, positions 1 or 4 are populated by serine or threonine. For instance, NhaP-II $K^+$-specific CPA1s are characterized by an $S_1T_2D_3A_4$ motif, where position 1 is populated by serine (Fig. 1). $K^+$-selective Kef-like CPA2s have an $L_1S_2S_3T_4$ motif, where position 4 is a threonine (Fig. 2). Having alanine and serine in positions 1-through-4 suggests a preference for small-size residues in this region in $K^+$-selective variants.

**Electroneutral EcNhaA triple mutant**. To experimentally test the computational analysis, we sought to rescue the inactive *E. coli* NhaA D163N mutant, in which $D_6$ of the CPA motif (Fig. 3a) was replaced with N. The D163N mutant was designed to convert the electrogenic EcNhaA to electroneutral, but, in essence, it does not grow under $Na^+/Li^+$ selective conditions, nor does it exhibit any $Na^+/H^+$ antiporter activity[40] (Supplementary Fig. 4a). Our evolutionary analysis provides a possible functional explanation for the mutant phenotype, suggesting multiple mutations that may rescue the D163N mutant. We generated a P108E_A160S_D163N triple mutant based on the following considerations. First, disruption of the electrostatic interaction between D163 at positions 6 of the CPA motif and K300 at position 8 yielded an unpaired lysine with a lone positive charge in the core domain of the D163N mutant. Based on the phylogeny and homology modeling of the human NHA2 transporter, also a CPA2, a conserved glutamate on TM-3[1] emerged as an alternative salt bridge partner for K300 (Fig. 4). We, therefore, introduced a glutamate in TM-3 by mutating the corresponding position in EcNhaA (P108) to Glu (P108E; Supplementary Fig. 5). This glutamate should compensate for K300's lone positive charge, thereby stabilizing the core domain. Second, the D163N mutation also resulted in a loss of a carboxylate group, which could participate in ion coordination. To compensate for this loss, we mutated A160 to serine (A160S; Supplementary Fig. 5), which would enable the hydroxyl group to participate in ion coordination, as proposed for the CPA1 member PaNhaP[8].

Cells of the *E. coli* mutant EP432 lacking the $Na^+/H^+$ antiporter genes *nhaA* and *nhaB*[42] were transformed with plasmids containing either WT or the triple mutant EcNhaA, or an empty pBR322 vector as a negative control. The transformants' growth on selective media (0.6 M $Na^+$ at pH 7 or pH 8.3 and 0.1 M $Li^+$ at pH 7) and antiporter activity in isolated everted membrane vesicles were then measured using a

quenching/dequenching assay, as described in Methods section (Supplementary Fig. 4a).

Cells transformed with both the WT and the triple mutant plasmids grew in the selective media at neutral pH, but the latter failed to grow at alkaline pH (Fig. 5). *E. coli* survival under alkaline conditions (pH > 7.6) is intrinsically dependent upon electrogenic transport. A net proton influx, resulting from the $1Na^+:2H^+$ stoichiometry, is required to maintain a constant internal pH in external alkaline conditions. Under such conditions, the ΔpH across *E. coli* membrane collapses, leaving only delta potential (ΔΨ)[43]. Electrogenic NhaA can utilize ΔΨ but an electroneutral variant cannot. Thus, the poor growth of cells carrying the triple mutant plasmid under alkaline conditions provides an indirect evidence for electroneutral transport.

As previously observed, WT NhaA antiporter activity with $Na^+$ was pH dependent, reaching a maximum of around 95% dequenching at pH 8.5. Similar maximal dequenching was observed with $Li^+$, but the pH dependence was less pronounced at the measured pH range (Supplementary Fig. 4b). The triple mutant activity with $Na^+$ was also pH dependent, but reached a maximum of 37% dequenching, as compared to the WT (Supplementary Fig. 4c). Significant but lower activity was observed with $Li^+$ (Supplementary Fig. 4c), while no activity was observed with $K^+$. Notably, because high copy number plasmids were used for the triple mutant, the expected expression level was way above that of a single copy chromosomal gene, which maintains the salt resistance phenotype. The triple mutant's apparent $K_m$ for sodium (0.25 ± 0.02 mM) was similar to that of WT EcNhaA, but that of $Li^+$ was almost 20-fold higher (0.41 ± 0.07 mM). This low affinity could be explained by the loss of the carboxylate group in the binding site, resulting from the D163N substitution, which may result in a less favorable coordination of the smaller lithium ions.

Electrogenicity vs. electroneutrality of the antiporter activity can be inferred directly from the effect of selectively abolishing the membrane potential component of the driving force[44]. In everted membrane vesicles, this is commonly accomplished by using the potassium ionophore valinomycin together with $K^+$. Under such conditions, the proton gradient is the driving force and when the ΔΨ collapses, ΔpH is increased. The net translocation of positive charge in electrogenic antiporters produces a positive out ΔΨ in the everted system that limits the rate of the antiporter activity. By dissipating this charge, the presence of valinomycin/K+ should accelerate the antiporter's rate[45]. In contrast, this manipulation should not affect the rate of an electroneutral antiporter.

Figure 6 shows the antiporter activity of the WT and the triple mutant in isolated everted membrane vesicles, in the presence and absence of valinomycin/K+. The collapse of the membrane potential has a drastic effect on the rate of the WT but hardly any effect on the rate of the triple mutant. Taken together, our results confirm our predictions; the newly introduced mutations A160S and P108E revived the dead mutant D163N, and the triple mutant is electroneutral, as expected.

## Discussion

We describe the phylogeny of the large and diverse CPA superfamily as a tree that captures the partition of the superfamily's two major clades and distinguishes between different phenotypes. We show that CPAs share eight highly conserved specificity-determining residues. These cannot be described as a linear motif, because they reside on different helices and do not cluster sequentially. However, they are spatially close to each other in the protein core and cluster around the substrate binding site, supporting their consideration as a motif (Fig. 3). Importantly,

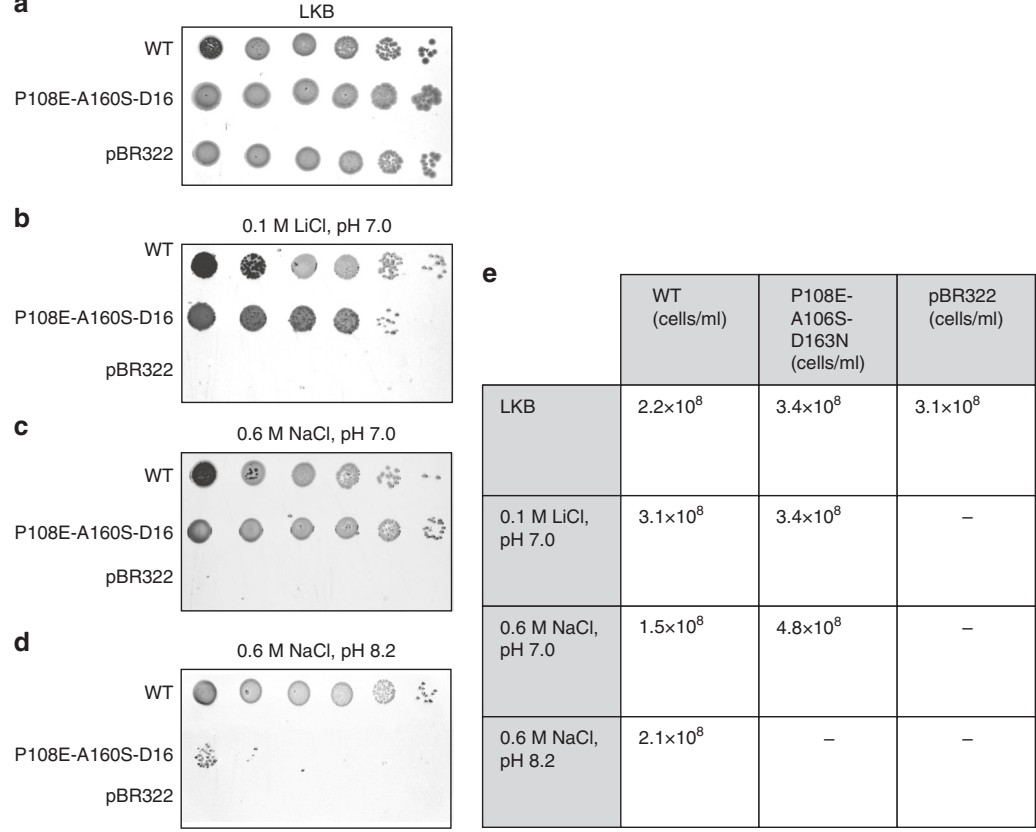

**Fig. 5** The NhaA triple mutant supports cell growth under neutral but not alkaline pH. EP432 *E. coli*, carrying deletions of both *nhaA* and *nhaB* genes, was transformed with EcNhaA, either WT or the P108E_A160S_D163N triple mutant, or an empty pBR322 vector, as negative control. Growth resistance to Na⁺/Li⁺ was performed as described in Methods section. **a** Cells were grown in modified L broth (LBK, with NaCl replaced by KCl) and then **b**–**d** tested for growth resistance to Li⁺ and Na⁺. **e** A summary of the number of cells grown under each condition. The standard deviation was 3–5%

variations of this motif can be used to distinguish between CPA1s and CPA2s, and affect functional characteristics such as electrogenicity and ion selectivity. This motif allowed us to understand the inactive phenotype of the EcNhaA D163N mutant and to successfully introduce two secondary site mutations that rescued it. The searchable database CAPTCHER (http://bental.tau.ac.il/CAPTCHER) maps any query CPA onto the phylogenetic tree, lists the CPA motif, and predicts its electrogenicity and ion selectivity characteristics.

Our results indicate that CPAs manifest four sequence variations involving positions 5 and 8, or 6 and 8 of the CPA motif that may provide stabilizing electrostatic interactions between TM-5 and TM-10 (Fig. 7). The nature of this interaction varies and distinguishes CPA1s from CPA2s. In CPA1s, position 5 of the CPA motif (TM-5) is populated with glutamate and position 8 is populated by an arginine (TM-10). As demonstrated in the crystal structures of PaNhaP1 and MjNhaP[7,8], these two position could salt bridge at least in some of the protein's conformations (Figs. 1, 7a). The high free energy penalty associated with introducing titratable residues into the hydrophobic core of membrane proteins suggests that this conserved interaction facilitates structural and/or functional roles. In CPA1s, neither the glutamate (position 5) nor the arginine (position 8) participate in ion coordination directly[8], and are unlikely to undergo protonation and deprotonation to facilitate proton transport, as required by the two proposed antiport mechanisms[13–15]. We therefore hypothesize that this interaction has mainly a structural role. These two positions are located at opposite ends of the core domain, and their interaction can contribute to its structural

rigidity. This, in turn, could ensure that throughout the conformational changes associated with transport, the core domain moves as a coordinated bundle.

In CPA2s, the putative interaction between TM-5 and TM-10 involves position 6 and 8 of the CPA motif and may be based on a salt bridge or hydrogen bond(s) (Figs. 2, 7b, d). For instance, in the crystal structures of EcNhaA[14] and TtNapA[6], D₆ (TM-5) salt bridges K₈ (TM-10; Fig. 7b). AtCHX17 and PaKefB, on the other hand, feature polar residues at position 6 (TM-5) and lysine at position 8 (TM-10; Fig. 7c), implying that potentially even the weaker hydrogen bonding interaction between these two residues is sufficient to maintain stability.

In the inactive EcNhaA D163N mutant (D₆, TM-5), the mutation perturbs the conserved salt bridge, leaving K300 (K₈, TM-10) as a lone positive charge in the protein core. This is expected to have a twofold effect: the loss of a favorable polar interaction holding the two helices, and the unfavorable desolvation of K300, which may induce conformational changes. By adding two second-site mutations, we successfully rescued D163N (Fig. 5; Supplementary Fig. 4). One mutation was P108E on TM-3, designed to salt bridge with K300, while retaining the hydrogen bond between N163 and K300 to maintain the TM-5-to-TM-10 link. The second mutation, A160S, was intended to provide the hydroxyl group of S160 to participate in ion coordination as proposed for PaNhaP[8] (Supplementary Fig. 5). Beyond the successful activity rescue, the growth experiments and the insensitivity of the triple mutant's to valinomycin (Figs. 5, 6) suggest that it is electroneutral, as opposed to the electrogenic WT EcNhaA, thus achieving the intended goal of the single mutant design of D163N.

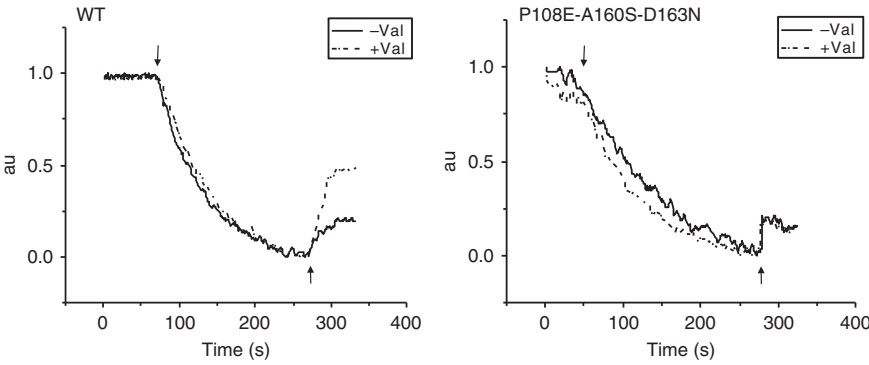

**Fig. 6** Valinomycin effect on $Na^+$-induced proton efflux confirms electroneutrality. For measuring the $Li^+/H^+$ antiporter activity in pH 8.5, *E. coli* EP432 cells expressing WT or the P108E-A160S-D163N mutant were grown in LBK (pH 7.0) and everted membrane vesicles were isolated. The $\Delta pH$ across the membranes was determined using acridine orange, a fluorescence probe of $\Delta pH$. The reaction mixture (2.5 mL) was modified according to Schuldiner and Fishkes[45] to get the maximal effect of valinomycin (2 μM). It contained, protein (100 μg), Tris-Cl (10 mM, pH 7.8), $MgSO_4$ (10 mM), and KCl ($Na^+$ free) 0.14 M. After a 1-min incubation, 9-amino- acridine (0.5 μM) was added. The $\Delta pH$ was generated by the addition of ATP (2 mM). Where indicated, 2 μM of an ethanolic solution of valinomycin was added 1 min prior to the ATP. At the onset of the reaction, D-lactate (2 mM) was added (downward facing arrow) and the fluorescence quenching was recorded until a steady-state level of $\Delta pH$ (100% quenching) was reached. Then, 10 mM NaCl was added (upward facing arrow), and the new steady state of fluorescence obtained (dequenching) was monitored. Whereas valinomycin/$K^+$ had no effect on the $Na^+$-induced dequenching by the mutant, it increased it dramatically in the WT. The experiments were repeated at least three times with practically identical results

Electrogenic versus electroneutral transport is one of the main functional characteristics of CPAs. There are two conflicting hypotheses regarding the mechanism for electrogenicity. One argues that $D_6$ is the second proton carrier in electrogenic antiporters, whereas the other points to $K_8$[13–15].

Based on the phylogenic analysis, existing experimental data and the data presented here, we propose a model for electrogenicity in CPAs. Our model requires two acidic residues that undergo protonation and deprotonation at positions 6 and 7 of the CPA motif (Fig. 7b), and a basic residue at position 8. The phylogeny together with experiments suggest that while position 7 could be populated by either aspartate or glutamate[46], an aspartate is preferred at position 6 and a lysine at position 8. As evidence, when $D_6$ is replaced with glutamate in EcNhaA the transporter is inactive[46]. When $K_8$ is mutated to arginine, EcNhaA is still electrogenic but activity decreases[13]. Indeed, we could not find such naturally occurring electrogenic sequence variants. Similarly, when $K_8$ was replaced with histidine in TtNapA, the transporter retained its electrogenic activity[24]. In contrast, when $K_8$ was mutated to glutamine, TtNapA was rendered electroneutral[24]. However, some CPA2s with a lysine at position 8 are suspected to be electroneutral as well[34]. Together with our triple mutant, the above experiments suggest that an aspartate in position 6 is crucial for electrogenicity and while any basic residue at position 8 is beneficial, it does not confer electrogenicity by itself. A possible explanation is that $K_8$ contributes to the dielectric environment, which in turn is important for modulating the pKa of $D_6$, the actual proton carrier. The preference of lysine over arginine and histidine in electrogenic CPAs is a matter for speculation.

In one plausible scenario for the electrogenic antiport cycle, the first $H^+$ would be released to one side of the membrane from $D_7$ upon $Na^+$ binding. Then, $D_6$ would also deprotonate and bind to the same $Na^+$, while its negative charge is also masked by $K_8$'s positive charge. Ultimately, $Na^+$ would be released at the other side of the membrane, and both aspartates would be protonated. The P108E_A160S_D163N EcNhaA triple mutant lacks an acidic residue at position 6 of the motif, and our indications that the triple mutant is electroneutral further support our model. P108E was inserted in a position similar to that observed in HsNHA2

(Fig. 4) to neutralize the unmasked charge of K300, while A160S was placed to coordinate the cation.

Consistent with our electrogenicity model, proteoliposomes of AtKEA2, a CPA2, mediate electroneutral transport[34]. AtKEA2 features lysine at position 8, but, as the model predicts, it also lacks the acidic residue at position 6 (TM-5), featuring glutamine instead. As glutamine cannot undergo protonation and deprotonation, electroneutral behavior is to be expected. Collectively, our results are more consistent with the traditional antiporter mechanism, where the acidic residue at position 6 is the second proton carrier[5,13]. However, the experimental data is limited to prokaryotic CPA2s and future experiments will be required to explore its validity to eukaryotic CPAs. It is not implausible that eukaryotic CPA2s function under different mechanism and that more than one mechanism for electrogenicity exists in this large and diverse family.

CPA2 mammalian NHAs, for example, seem to diverge from this model. Though characterized by two aspartates at positions 6 and 7 and an arginine at position 8 (Fig. 7d), they are electroneutral[24]. To address this inconsistency, we searched for sequence features that set this clade apart from electrogenic CPA2s. We identified a highly conserved glutamate (E215) that is unique to this group. A homology model of HsNHA2 revealed that this glutamate, on TM-3, could salt bridge with $R_8$ of the CPA motif (Fig. 4). This slight change in the dielectric environment of $D_6$ could, theoretically, prevent protonation and deprotonation from occurring. Therefore, only one proton would be transported, rendering this clade electroneutral.

The phylogenetic analysis further shows that the phenotypic electroneutral/electrogenic partition correlates with the CPA1/CPA2 division only with respect to electroneutral CPA1s (Fig. 7; Supplementary Fig. 7b, c). In CPA1s, the conserved aspartate at position 7 of the motif is expected to be both the ion and proton carrier. The asparagine at position 6, the glutamate at position 5 and the arginine at position 8 are unlikely to undergo protonation and deprotonation to facilitate the transport of a second proton (Fig. 7a). Thus, all CPA1s are electroneutral. However, not all CPA2s are necessarily electrogenic, as this phenotype necessitates two acidic residues at positions 6 and 7 and a basic residue at position 8 (Fig. 7b). CPA2s that lack either of these two features

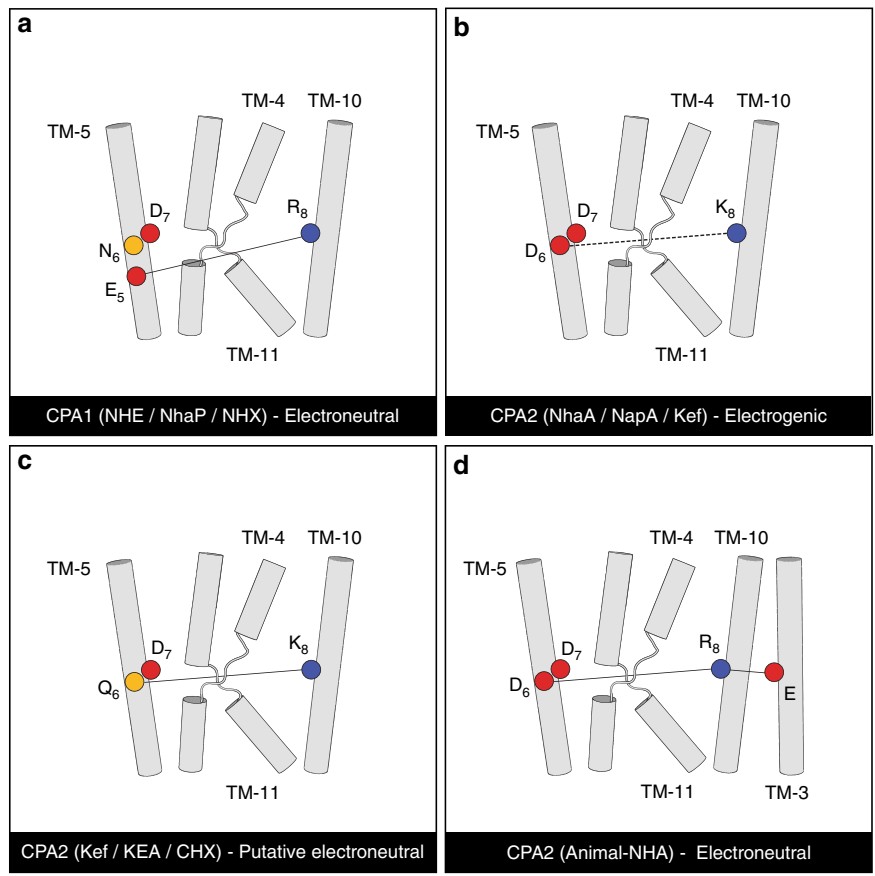

**Fig. 7** A proposed model of electroneutrality versus electrogenicity. The four main variations of the putative interaction between the conserved polar residues on TM-5 and TM-10 are shown. Helices are numbered according to EcNhaA's topology. The relevant residues are shown as circles, with acidic in red, basic in blue, and polar in yellow. Interactions that are expected to remain intact throughout the transport cycle according to the proposed model are marked with solid lines in **a**, **c**, **d**, and the interaction that would alternate is marked with a dashed line in **b**. **a** The electroneutral CPA1 subtree would feature a salt bridge between arginine in position 8 and glutamate in position 5. The protonation state of these residues would be fixed, while that of $D_7$ would change upon proton transport. **b** Electrogenic CPA2s, characterized by acidic residues in positions 6 and 7, presumably interacting with the two protons, and a basic residue in position 8. Upon interacting with a proton, $D_6$ would alternate between salt bridging and hydrogen bonding with $K_8$. **c** Putative electroneutral transport by CPA2s that lack the acidic residue in position 6. Positions 6 and 8 would hydrogen bond with each other. Similar to electroneutral CPA1s, the protonation state of these residues would be fixed, while that of $D_7$ would alter upon interaction with the proton. **d** Electroneutral mammalian NHA-like CPA2s, surprisingly featuring two acidic residues at positions 6 and 7, similar to the electrogenic CPA2s shown in **b**. This group also includes an arginine at position 8 that is prevalent in electroneutral CPAs, and a uniquely conserved glutamate on TM-3 that potentially could salt bridge with each other. This slight change in the dielectric environment of $D_6$ could, theoretically, prevent its protonation and deprotonation, resulting in electroneutral transport

are expected to be electroneutral (Fig. 7c, d). Thus, electrogenicity either evolved independently in different clades of the CPA2 subtree, or evolved once and was lost during evolution in multiple occasions.

Another important feature that divides the CPAs according to the phylogenetic analysis is the unwound section of the ion binding site (TM-4 in EcNhaA). Comparing $K^+$-selective to $Na^+$-selective clades, we predict that small changes in this region may alter cation selectivity (Fig. 3c). For example, in the structure of the $Na^+$-selective CPA1 PaNhaP, a thallium ion presents a coordination pattern of five interactions, contributed by the backbone and side-chains of four amino acids[8] (Fig. 8e). According to the phylogenetic analysis, at least three of these residues are conserved across all CPA1s. Hence, as suggested, $Na^+$ follows a similar coordination pattern in other CPA1s as well. Potassium is slightly larger than sodium, and has a larger coordination number, often six interactions. Consistent with this our analysis shows that CPAs that are expected to be $K^+$-selective

have an extra polar residue in their binding site on the unwound section of TM-4. This residue, usually serine or threonine, may contribute the extra coordination needed for potassium ions (Fig. 8f). Indeed, introducing a polar residue to position 4 of the CPA motif in the $Na^+$-selective *Zygosaccharomyces rouxii* SOD22 transporter, improved its tolerance to KCl stress[16]. Moreover, the smaller residues characterizing the unwound section of TM-4 in a large portion of the $K^+$-selective CPAs may further contribute to the correct coordination of the cation by slightly enlarging the cavity. In light of this, we predict an A130S_P133A MjNhaP double mutant to be $K^+$-selective (Fig. 8).

Presumably, CPAs may be characterized by additional phenotypes, such as transport rate, substrate affinity, and possibly also the presence of allosteric binding sites that are involved in regulation pathways. These may also be reflected in the various clades of the phylogenetic tree. For experimental characterization, the approach presented here can be readily extended to finding specificity determinants linked to other phenotypes. Such studies

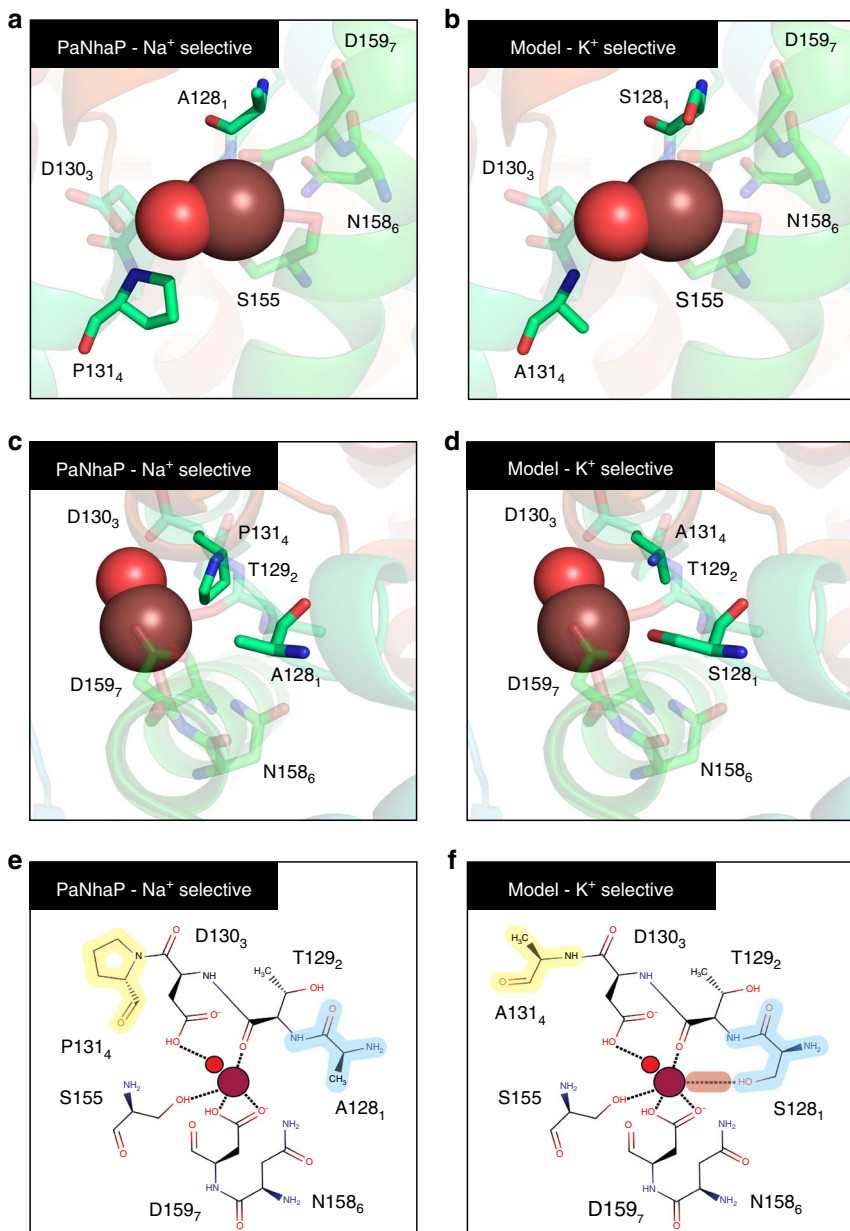

**Fig. 8** A proposed model for potassium-selectivity in CPA1s. Side view (**a**, **b**), top view (**c**, **d**), and schematic two-dimensional projection (**e**, **f**) of the binding site of the $Na^+$-selective PaNhaP (PDB id 4CZA). **a**, **c**, and **e** are the wild-type structure and **b**, **d**, and **f** are a model structure of a proposed PaNhaP mutant, which we suggest to be $K^+$-selective. Residues that participate in ion coordination are shown as sticks. The positions where we suggest mutations are in bold. A thallium ion (bronze) and a water molecule (red) are shown in the binding site. **e** vs. **f** The suggested additional coordinating interaction between the protein and the ion and the slight increase in the pore diameter resulting from the A128S (light-blue) and P131A (light-yellow) mutations, respectively, are shown. The A128S substitution is expected to provide another polar interaction with the ion, consistent with the expected larger coordination number of potassium ions compared to sodium. The P131A substitution is expected to slightly increase the cavity to accommodate the larger potassium ions **a** vs. **b**

might lead to the expansion of the motif beyond the eight positions introduced here. In principle, our approach is also applicable to other protein families, which are diverse in terms of sequence and phenotypes, and where structural knowledge is available. Obvious candidates include the apical sodium-dependent bile acid transporters (ASBTs), leucine transporters (LeuT), and sodium-coupled citrate transporters, whose structures resemble the CPAs.

The observed activity of the rationally designed triple mutant, in spite of involving radical changes, including a Pro-to-Glu replacement in the most sensitive region of EcNhaA, demonstrates a clear understanding of the delicate electrostatic balance

at a level that allows manipulation. It also confirms the value of such fundamental understanding, revealing intriguing implications on drug design based on disease-associated human homologs.

## Methods

**Sequence search and clustering**. Protein databases include tens of thousands of sequences predicted to be cation/proton antiporters. In order to produce a representative set of sequences for the entire superfamily, we implemented the following steps. First, we searched the Swiss-Prot database[47], which contains high quality manually curated sequences, for entries denoted as CPAs (CPA1 and CPA2 members) that share the NhaA-fold. This initial search yielded a total of 328 proteins. Redundant proteins were filtered using CD-HIT-v4.6.6[48] with a threshold

of 90% sequence identity, obtaining 146 representatives. A seed MSA was constructed by aligning the membrane segment of these using MAFFT v7.222 with default parameters[49].

Using HMMER-3.1[25] and the seed alignment we produced an HMM profile and used it to broaden the search against the UniProt[50] reference proteomes database at 55% co-membership threshold (RP55; http://pir.georgetown.edu/rps/ released in November 2016). We selected RP55 because it was found to best follow the common taxonomic classifications[51]. We discarded fragmented or uncharacterized sequences, as well as short sequences (<300 amino acids) that appeared too short to truly represent the full NhaA-fold. This resulted in a pool of 15,252 proteins.

To reduce the size of this compilation, we clustered the proteins using CD-HIT with different thresholds (50%, 55%, 60%, 65%, 70%, and 75% sequence identity), leaving one sequence of each cluster as representative. The 70% threshold reduced the number of sequences by almost half while preserving the taxonomic classification integrity within each cluster in the phylum level. That is, in more than 99% of all clusters, all the proteins in the same cluster were from the same phylum. The scaled-down pool consisted of 7772 proteins.

**Multiple sequence alignment.** Producing a multiple sequence alignment for thousands of sequences is challenging, especially when distantly related proteins are included. We thus exploited hmmalign that aligns each protein from a pool to a given HMM profile. Using hmmalign, we aligned the 7772 proteins to the HMM profile produced from the seed alignment (see above). We then screened out proteins that, according to the alignment, lacked critical functionally important helices, which are indispensable for CPAs activity, such as TM-5 of EcNhaA that harbors the ion binding site. The final MSA consisted of 7499 proteins. While the resulting MSA contained many indels, gap-rich regions in the MSA were not removed in light of the findings of Dessimoz and colleagues[52], which showed that on average trimmed alignments tend to produce inferior phylogenetic trees.

**Phylogenetic tree reconstruction.** We used two different methods for phylogeny reconstruction. A fast approximation of the (unrooted) tree was built using FasTree 2[26]. FastTree uses heuristic neighbor-joining for building an initial topology. It then maximizes the tree's topology and branch lengths by using maximum likelihood rearrangements[26]. The analysis was performed using the Le and Gascuel amino acid replacement model (LG model)[53] and discrete Gamma model with 20 rate categories. Bootstrap analysis was performed using 100 replicas, produced with PHYLIP-3.695 SEQBOOT program[54] and FastTree 2. Bootstrap values were projected onto the tree topology inferred based on the base MSA.

In addition, we used the more robust IQ-TREE-1.6.2 algorithm[27], which combines the hill-climbing approach and a stochastic perturbation method. This algorithm allows for better search of the optimal maximum-likelihood tree compared to heuristic approaches. To determine the best substitution model that fits the data, we applied IQ-TREE's model finder tool[55] including mixture models to the full list of representatives. The LG substitution model with empirical amino acid frequencies calculated from the data and a FreeRate model with 10 categories was predicted to best fit the data. Bootstrap analysis with 100 replicas was performed using IQ-TREE and bootstrap values were projected onto the tree topology inferred based on the base MSA.

Next, RogueNaRok[56] was utilized to identify rogue sequences that reduced the bootstrap support values of the inferred maximum likelihood tree. By default, RogueNaRok searches for detrimental taxon one by one. Nevertheless, there may be groups of several taxon that should be pruned together (dropsets) in order to observe improvement in the tree's support. RogueNaRok's "-s" parameter allows such option, however for our large dataset a search for dropsets larger than two sequences became computationally highly expensive. Thus, at first, we invoked RogueNaRok with –s equals 2, and removed the identified rogue sequences as they all had an impact on the bootstrap value of at least one of the main clades of the tree. A main clade was defined as the largest clade with a relatively high bootstrap (60%) that is close to the tree's base. In order to estimate the effect of dropsets larger than two on the support of these main clades, we color-coded each clade and visually detected dropsets that were also removed. Dendroscope-3.5.8[57] was used for tree visualization. Potentially, however, there are other dropsets that affect the support values within each of these clades. The final phylogenetic trees consisted of 6537 and 6597 sequences with 1990 and 1829 aligned sites that were analyzed for the IQ-Tree and FastTree analyses, respectively.

To further assess the robustness of our results, we performed an additional IQ-TREE analysis on the 500 most divergent sequences, detected using the phylogenetic diversity analysis tool PDA-1.0.3[29]. IQ-TREE was then utilized for tree reconstruction and bootstrap analysis, as described above.

Overall, all three phylogenetic trees reproduced the same main CPA clades with minor differences (Supplementary Figs. 3, 8), as described in the main text and supplementary notes. One exception though is the GerN clade that was assigned an extremely low bootstrap value of only 10% in the scaled-down analysis of the 500 most divergent sequences. Some sub-clades of the GerN branch are characterized by relatively high bootstrap values, and we believe that the overall low support might result from our failure to identify all dropsets.

**Homology modeling of HsNHA2.** We used HHpred[58] to find structural templates for HsNHA2. The archaeal CPA1 member MjNhaP1 (PDB ID 4CZB) and the bacterial CPA2 member TtNapA (PDB ID 5BZ2 and 5BZ3) were identified as the best templates, with sequence identity of 19% and 16%, respectively. As the two templates show high structural similarity (TM-score of 0.85 and RMSD of 2.98 Å calculated for 4CZB and 5BZ2 by TM-align[59]), and because HsNHA2 was assigned to the CPA2 clade, we chose TtNapA as a template. Pairwise alignments that assign each position in the query sequences to the corresponding position in the template structures were deduced from the MSA used for the phylogenetic tree. We then used MODELLER 9.18 (with default parameters)[60] and the pairwise alignment to build an ensemble of 100 models based on the inward-facing structure of TtNapA (PDB id 5BZ2) and 100 other models based on its outward-facing structure (PDB id 5BZ3). All models were then refined using short steepest descent energy minimization with GROMACS-5.1 and the AMBER99SB-ILDN force field[61,62]. To estimate whether E215 could salt bridge with R432 we calculated the distance between the two residues in all 200 models, which represent different side chain rotamers. In 92% of the models based on the inward-facing template and 79% of the models based on the outward-facing template, E215 and R432 were at a distance lower than 4 Å and thus, could potentially salt bridge with each other.

**Modeling the EcNhaA P108E_A160S_D163N triple mutant.** The EcNhaA triple mutant model was produced using the WT crystal structure (PDB id 4AU5). Using PyMOL-2.0.6[63] we replaced P108 with Glu, A160 with Ser, and D163 with Asn. While the A160S and D163N substitutions are moderate in terms of the amino acid size, the P108E mutation is not. We thus produced 15 models with all the possible side-chain rotamers predicted for E108 by PyMOL. For each model, a short steepest descent energy minimization was performed with GROMACS and the AMBER99SB-ILDN force field[61,62]. The model with the lowest predicted energy was then selected.

**In silico model validation.** We used ConSurf[36] (with default settings) to examine the evolutionary conservation patterns of the HsNHA2 model. Homologs with 35–95% sequence identity were collected using HMMER with one iteration and an E-value of 0.0001 against the clean UniProt database. 120 unique sequences were retrieved, and MSA was produced using MAFFT-L-INS-i. The rate of evolution at each site was calculated using the Bayesian method and the LG evolutionary model. The expected pattern is that the core of the protein would be highly conserved and the periphery more variable. Overall, the helices assignment followed the expected evolutionary pattern with the exception of the 7th helix (corresponds to TM-6 in EcNhaA), where the assignment was moved by one amino acid. To further examine the homology models, we colored their amino acids according our in-house hydrophobicity scale, and showed that the lipid-exposed residues are hydrophobic, while the core is more polar (Supplementary Fig. 6).

**Defining the CPA motif.** A number of supervised classification methods specialize in identifying sequential motifs that segregate proteins into different classes using MSAs. These methods require that each sequence is annotated by a certain feature. In our case, for example, such features might be CPA1s vs. CPA2s, electrogenic vs. electroneutral transporters, and $Na^+$- vs. $K^+$-selective ones. The accuracy of the results will thus depend on the right assignment of a feature to each of the sequences. However, when it comes to CPAs, the assignment of the above features to a given protein is often missing or debatable. In the UniProt database, most CPAs are not assigned with an electroneutral or electrogenic transport mechanism. Moreover, in several cases, proteins annotated as $Na^+$-selective show higher degrees of similarity to proteins annotated as $K^+$-selective, and vice versa, undermining the accuracy of these annotations. Thus, we applied the curated approach described below for identifying the sequential features that discriminate between different CPA classes.

We analyzed the reduced data set of the 500 most divergent sequences using ConSurf[36]. The MSA containing these 500 sequences, together with the phylogenetic tree reconstructed with IQ-TREE, were given as an input. The rate of evolution at each site was calculated using the Bayesian method and the LG evolutionary model. We focused on highly conserved residues that scored 8 or 9 in the ConSurf conservation scale (1 being the most variable and 9 the most conserved). As anticipated for a helical membrane protein, a large portion of these amino acids were either the small glycine and alanine residues or the bulky and hydrophobic leucine, isoleucine and valine[37]. Glycine and alanine facilitate close helix packing, while leucine, isoleucine and valine, and occasionally also alanine, are favored in the hydrophobic environment of the membrane and the protein core. Instead, in order to analyze function and the transport mechanism, we focused on polar and charged residues, identifying 46 positions of potential interest. We then plotted the amino acids distribution for each position (Fig. S9), looking for those with relatively low diversity. Projecting this data onto the phylogenetic tree, we searched for correlation between these conserved sequential features and some of the CPA superfamily's characteristics, such as electrogenicity and ion selectivity. We also used the available CPA structures to guide us in selecting positions that are more likely to have a direct role in the transport mechanism. Such residues are expected to be in the protein interim or close to its two discontinuous funnels, rather than in the periphery.

By far, position 7 in the CPA motif, populated almost exclusively by aspartate (94% of the sequences), emerged as the most conserved and homogenous position, and was therefore chosen as the first amino acid in the CPA motif. This aspartate corresponds to Asp-164 of EcNhaA, known to be crucial for ion coordination[40]. Another conserved position was position 8 of the CPA motif (position 300 in EcNhaA). Forty-seven percent of the proteins featured arginine in this position, while 48% featured lysine. Thus, at first, this position emerged as a perfect candidate for distinguishing CPA1s from CPA2s, represented by the two main branches of the tree. However, projecting the amino acids distribution onto the tree, the lysine/arginine duality did not perfectly match the CPA1/CPA2 division (Supplementary Fig. 7a). However, the coincidence of both arginine at position 8 and a glutamate at position 5 of the CPA motif (residues 300 and 159 in EcNhaA, respectively) resulted in a perfect match (Supplementary Fig. 7b). Thus, these two positions were added to the motif as well. Next, position 6 (residue 163 in EcNhaA), that correlates to the first position of the ND/DD motif[12], showed more diversity than anticipated. Nevertheless, projecting its distribution onto the tree, together with the already established CPA1/CPA2 marker, revealed that aspartate in this position and lysine at position 8 correlated to transporters shown to be electrogenic relatively well (Supplementary Fig. 7c). Based on the proposed model for electrogenicity described in the main text, we later broadened the definition of an electrogenic CPA. We hypothesized that this class of CPAs includes any transporter with a lysine at position 8 of the CPA motif and either aspartate or glutamate at position 6. Finally, positions 1-through-4 of the CPA motif (residues 131-through-134 in EcNhaA), correlating to the unwound section of TM-4, enabled distinction between Na+- and K+-selective CPAs. More specifically, most K+-selective CPA1s were characterized by serine at position 1 and alanine at position 4 (Supplementary Fig. 7d). In K+-selective CPA2s, position 2 and 3 were characterized by two small residues, serine and alanine, while in most cases, position 4 featured the polar threonine.

In total, these eight conserved positions make up a minimal set that enables discrimination between different types of CPAs. That is, between CPA1 and CPA2 members, electroneutral and electrogenic CPAs, and Na+- and K+-selective CPAs. Looking at the amino acids distribution, there are other positions that emerged as highly conserved. These include, for example, R331 and E333 in TtNapA (G338 and T340, respectively in EcNhaA), both located at the unwound section of TM-11. R331 is highly conserved in CPA1 members compare to CPA2s. As much as 90% of CPA1s are characterized by a basic residue in this position, compare to roughly 30% of CPA2s. Thus, the distinction between CPA1s and CPA2s cannot be attributed to this position. In CPA2s, such a basic residue could be found in electrogenic NapA-like but not NhaA-like transporters. Since CPA1s that feature a basic residue in this position, are electroneutral, it follows that it probably does not confer electrogenicity. Finally, this basic residue could be found in both Na+- and K+-selective CPA1s and thus does not determine ion selectivity (Supplementary Fig. 7e). As for E333, although conserved in some CPA2s (Supplementary Fig. 7f), it is unlikely to mediate proton transport as it cannot explain the existing experimental data. Drew and colleagues demonstrated that the replacement of K305 with glutamine in TtNapA resulted in an electroneutral transport[24]. While this could, in theory, imply a direct involvement of K305 in electrogenic transport, as the proton carrier, it may also be the result of an indirect action of K305. That is, K305 may act by modulating the dielectric environment of an adjacent titratable residue, which is the component that determines electrogenicity. Our experimental data support the indirect involvement of K305. As K305 is ~12 Å apart from E333 in both the inward- and outward-facing conformations of TtNapA, we find it hard to believe that the latter could participate in proton transport directly or indirectly. Located close to the ion binding site, E333 could potentially participate in ion coordination, however, as it can be found in both Na+- and K+-selective CPA2s it does not confer ion selectivity and was not included in the motif.

**Bacterial strains and plasmids.** EP432 is an *Escherichia coli* K-12 mutant[42] (*mel*BLid, Δ*nhaA*1::*kan*, Δ*nhaB*1::*cat*, Δ*lacZY*, *thr*1) constructed from bacterial strain NM81[64]. EP432 cells do not grow on high Na+ (0.6 M at pH 7 or pH 8.3) or high Li+ (0.1 M at pH 7) selective conditions, and do not exhibit Na+/H+ antiporter activity, unless transformed with a plasmid expressing an antiporter similar to NhaA. Cells were grown in either L broth (LB) or modified L broth (LBK; with NaCl replaced by KCl)[64]. The medium was buffered with 60 mM 1,3 bis-{tris (hydroxymethyl)-methylamino} propane (BTP). For plates, 1.5% agar was used. Antibiotic was 100 μg/mL ampicillin. The pAXH3 plasmid encodes wild-type (WT) His-tagged NhaA[65]. The D163N mutant was previously described[40,46]. Mutagenesis of the triple mutant was performed by site-directed mutagenesis using a PCR-based protocol with pAXH3 as a template (Supplementary Table 1). The *nhaA* gene, including the newly introduced mutations, was sequence-verified. To test growth resistance to Li+ and Na+[66], cells were grown on LBK to OD600 of 0.6–0.7. Samples (2 μL) of serial tenfold dilutions of the cultures were spotted onto agar plates containing the indicated concentrations of NaCl or LiCl at the various pH values and incubated for 48 h at 37 °C.

**Assay of Na+/H+ antiporter activity.** Everted vesicles from EP432 cells[42] transformed with the respective NhaA variants were isolated as previously described and used to determine Na+/H+ or Li+/H+ antiporter activity[67,68]. The antiporter activity assay was based on the measurement of Na+- or Li+-induced changes in the ΔpH as measured by acridine orange, a fluorescent probe of ΔpH. The fluorescence assay was performed in a 2.5 mL reaction mixture containing 50 to 100 μg membrane protein, 0.5 μM acridine orange, 150 mM choline chloride, 50 mM BTP, and 5 mM MgCl2, and pH was titrated with HCl. After energization with D-lactate (2 mM; Supplementary Fig. 4), fluorescence was quenched to achieve a steady state. Fluorescence quenching indicated that protons entered the vesicles upon energization (Supplementary Fig. 4a, down facing arrow). Subsequently, 10 mM of either Na+ or Li+ or K+ was added. A reversal of the fluorescence level (dequenching) indicated that protons have exited the vesicles in antiport with the respective cation (Supplementary Fig. 4a, up facing arrow). In this assay, a good estimate of the apparent $K_m$ for the cations is obtained from the cation concertation that yields 50% activity[45]. This concentration of Na+ hardly changed for the triple mutant as compared to the WT as discussed in the results section above. On the other hand, Supplementary Figure 4 shows that the $V_{max}$ has changed. Most importantly, as opposed to the WT of which rate changes dramatically in the presence of valinomycin/K+, the rate of the mutant does not change (Fig. 6).

## Data availability

The data supporting the findings of this manuscript are available from the corresponding author upon request. The searchable database CAPTCHER (http://bental.tau.ac.il/CAPTCHER) maps any query CPA onto the phylogenetic tree, lists the CPA motif, and predicts its electrogenicity and ion selectivity characteristics.

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

## Acknowledgements

This study was supported by a German-Israeli Project Cooperation (DIP) grant to NB-T and E.P. G.M. was funded in part by a fellowship from the Edmond J. Safra Center for Bioinformatics at Tel-Aviv University and the Madaim scholarship. M.D. was funded by the PBC, Council for Higher Education and the Hebrew University Program for Fellowships for Outstanding Postdoctoral Fellows from China and India. N.B.-T.'s research is supported in part by the Abraham E. Kazan Chair in Structural Biology, Tel Aviv University

## Author contributions

G.M., I.M., E.P., and N.B.-T. conceptualized the project. G.M. performed the phylogenetic analyses, evolutionary conservation analyses, and homology modeling; M.D., A.R., and Y.G.M. constructed the mutants and conducted the functional assays; H.A. and I.M. advised on the phylogenetic analyses; A.K. contributed to the discussion of the results

and all authors commented on and revised. The manuscript G.M., A.K., and N.B.-T. wrote the manuscript with input from all authors who reviewed the final paper.

## Additional information

**Competing interests:** The authors declare no competing interests.

