## [Peer Review File · Nature Communications]

Reviewers' Comments:

Reviewer #1:

Remarks to the Author:

This is a very timely paper addressing an important question: what residues are critical for electrogenic versus electroneutral transport of Na(K)/H⁺. Does a CPA exchange

1 C⁺: 1 H⁺ (electroneutral), or

1 C⁺: 2H⁺ (electrogenic).

CPA transporters mediate pH and cation balance in all living cells, and they impinge on many aspects of cell biology. For electroneutral exchange, the driving force is assumed to be from a pH gradient. However if the antiporter is electrogenic, then an electrical gradient could drive the antiporter. Thus efforts to differentiate these two transport modes is critical for a molecular understanding of biological processes.

EcNhaA has been extensively studied and has been the key model for understanding structure-function relationships of Cation/proton antiporters (CPA). Those studies formed the bases for several ideas:

i) EcNhaA was shown to be an electrogenic antiporter, which mediates 1Na⁺ for 2H⁺.

It contains a double DD in TM5 (D163, D164).

D164 is highly conserved in all CPAs and is the ion binding and translocation site alternatively for H⁺ and for Na⁺ in the transport cycle.

ii) The double DD in TM5 was postulated to mediate electrogenic Na/2H⁺ exchange, and the two DDs could serve as 2 H⁺ carriers.

iii) EcNhaA is phylogenetically classified in the CPA2 family which is separated from another CPA1 family. One hypothesis was that CPA2 members were electrogenic, while CPA1 mediated electroneutral antiport.

However, a recent paper by Uzdavinyis et al (2017 PNAS) provided experimental evidence that counters the former idea in (ii). They showed by mutagenesis and reconstitution of TtNapA, that a K305 is involved in electrogenic transport. They proposed that D157 and K305 of NapA (or D164 and K300 in EcNhaA) are two dominant H⁺ carriers.

To date, the molecular bases for the separation of CPA1 and CPA2 families has not be carefully examined. Here, Ben-Tal's group has collected over 6 thousand CPA sequences, and conducted a comprehensive analysis to determine which residues are critical for electrogenic versus electroneutral C/H⁺ antiport. They determined amino acids in 'conserved' motifs based on modeling CPAs which are predicted to assume a 'NhaA-protein fold' based on EcNhaA and 3 other crystallized structures.

Based on their analyses and mutagenesis experiment converting EcNhaA to an electroneutral antiporter, authors conclude that results support the proposed mechanism of where D(6) of the D(6)D(7) motif as the second H⁺ carrier and thus for electrogenicity.

Major Comments:

1. This is a very informative and useful study: a) to examine over 6 thousand sequences; b) to determine the molecular bases for separating CPA1 and CPA2, and c) to predict which residues mediate electrogenic exchange or not based on residues in the active motif and other expts. This is a very useful approach that is applicable to other transport proteins. As one can predict transport mode of poorly studied proteins based on active motifs of characterized homologs.

For transporters with high homology to EcNhaA, the conclusion would likely apply. Most prokaryote CPA2 have a DD motif whereas CPA1s have a -ND-. However, eukaryote CPA2s are mostly found in plants and some in fungi (Chanroj et al. 2012). CPA2 are rare or not found in most metazoan. HsNha2 is an exception.

2. So it is questionable whether this blanket statement applies to other CPA2 which do not have a DD in TM5? reasons are:

a) Most eukaryote CPA2 do not have a DD motif. Only –ND- or –XD- with a few exceptions: e.g. HsNha2 (R (8)), or Dicdi-Nhe3 (K(8)) have a –DD- motif.

If Masrati et al.'s proposal is right, then Dicdi-Nhe3 should be electrogenic, and HsNha2 could be mutated to become an electrogenic antiporter.

b) There is considerable variability in residues at the 6 position: –DD- (6,7).

I.e. –ND-, -OD-, -TD-, -SD-. Within several plant CHX families, variation suggests this residue is not critical for separating CPA2 from CPA1. Other residues likely work together with the X(6).

3. The most conserved residue among CPA2 is a K, Lysine. in TM10. Could K (8) form a salt bridge with another acidic residue (other than D (6))?

4. Additional core residues may exist in the TM11- unwound region. Authors do not consider/discuss residues here, though K383 was shown to affect activity in the CPA2-type CHX17, as monitored by growth of yeast mutants (Czerny DD. et al. 2016. BBA). A basic residue is conserved here as a K or R in several CPAs, e.g. TtNapA. It is likely a basic residue has some function?

5. There is an additional conserved acidic residue near the TM11-unwound region of CPA2, several prokaryote CPA2 show a glutamate, though an acidic residue is not seen in EcNhaA. Is it possible, EcNhaA may not be representative of some CPA2 with respect to residues critical for electrogenicity?

6. Proposed Experiment: The idea would be more convincing with additional experiment.

If the hypothesis of Masrati et al is correct, then HsNha2 (electroneutral) should be convertible to electrogenic antiport by converting R (position 8) to a K.

HsNha2 has a DD motif (TM5), but an R (8) instead of K.

Line 391. Authors propose this expt in the Discussion. Inclusion of this experiment would be critical.

Alternatively, show that transport of a plant CHX (CPA2) is electroneutral.

Other comments:

a) Fig. 2. Does the subclade CHX refer to mostly plant proteins?

If so, is the motif for D6-D7 in error? Most plant CHX genes encode –X6-D7-. If authors refer to prokaryotes only, then indicate.

This reviewer is not aware of –DD- which is probably very rare in eukaryote CHX TM5. Please revise.

b) Fig. S6. Method for measuring transport activity. Typos probably in reaction mixture regarding milliMolar versus microMolar. E.g. protein at 50-100 mg?

Acridine Orange at 0.5 mM.??

c) Line 144. Plant CHXwere. 'never shown to be electrogenic'. This statement is misleading. So far there has been no direct transport assays of any plant CHXs. So the possibility has yet to be tested. The statement might be revised to: 'Whether plant CHXs are electrogenic or not has yet to be established.'

d) Other differences between CPA1 and CPA2: Would authors discuss if the basis for different pH set points can be identified in the structure? There is E78 in TM2 EcNhaA which is conserved in many CPA2, but not in CPA1. If this is a pH sensor, how would it affect electrogenic versus electroneutral transport?

Reviewer #2:

Remarks to the Author:

The authors revisit the interesting question how to classify the sodium/proton antiporter proteins. They show that a newly identified CPA family motif is related to (1) the mode of transport (electrogenic vs electroneutral) and (2) ion selectivity.

Based on their evolutionary data and homology modeling they put forward a set of models for the structural correlates of electrogenic vs electroneutral transport and ion selectivity. They perform a functional study on the EcNhaA transporter and use their structural insights to rescue an otherwise dead mutant (D163N) that has been known for a long time. The new triple mutant restores transport and the authors suggest that it also changed from electrogenic transport to electroneutral.

Overall this is a paper that is likely to generate interest in the field of CPA transporters. The new hypothesis about the origin of the Na⁺ vs K⁺ selectivity suggests new experiments and computer simulations. A general issue with work based on sequence information is that correlation does not necessarily imply causation and although the authors perform structural modeling and one experiment, I still find that the weakest part of the paper is the attempt to draw inferences about molecular mechanisms from sequence data and static homology models.

(As the authors will without doubt realize when reading my review, I do have my own opinions about detailed mechanisms in the CPA family, but I do appreciate that their interpretation of their data is one possibility; I would just like to stress that there remain other possibilities, too, and I would suggest to be clearer about the limitations of interpreting the current data.)

Comments

- p3, L6: stoichiometry *could* be different from 1:1 or 2:1

- p4, L86

abstract: "CPA1/CPA2 division only partially correlates with electrogenicity"

The authors state that it has not been clear to many in the field of CPA transporters that the CPA1/CPA2 families do not strictly correlate with electroneutral or electrogenic transport. The paper makes an important contribution driving home this point.

I find it less convincing to specifically single out the paper by Brett et al 2005 (Ref 20) as having put forward this view. Although there is a sentence in the paper where the authors suggest a CPA2 transporter should be looked for electrogenic transport in crab

gills, they also state that, for example, in fungi, no electrogenic NHAs (CPA2) are known. So on balance, my reading of Brett 2005 is more in line with what the authors also propose (notwithstanding the fact that the present paper now places the fungal NHAs closer to the eukaryotic NHEs in the CPA1 clade than the animal NHAs in the CPA2 clade).

- p7, L158: "Because the tree can reproduce the CPA1/CPA2 division ..."

It is not clear from the main text that the fungal NHAs were re-assigned from CPA2 to CPA1 (Ref 20 considered then part of the CPA2 family). The supplementary text discusses this in more detail but is easy to overlook. Perhaps make clearer that the reassignment of fungal NHAs to CPA1 a new result of this paper.

It is therefore at least technically not true that "the tree can reproduce the CPA1/CPA2 division" because the authors change the CPA1/CPA2 division. Make clearer which results confirm existing knowledge and which ones update or challenge it.

- p8, L 206 "position 8 is populated by lysine (K300), that salt bridges with aspartate at position 6 (D163)"

The Lee et al (2014) [Ref 6] EcNhaA crystal structures show the mentioned salt bridge. However, there are inherent limitations to trying to derive a mechanism from static structures and supposing that the salt bridge always exists might not do justice to the actual molecular mechanism.

The "salt bridge breaking model" [Ref 6] proposed (based on MD simulations) that interactions with cations can induce localized changes that change the pKa of K300 substantially. These results were computationally validated recently [Y. Huang et al. Mechanism of pH-dependent activation of the sodium-proton antiporter NhaA. Nature Communications, 7:12940, 10 2016] where the local pKas were computed with one of the currently best pKa method (constant pH MD simulations). The paper shows how the breaking of the D163-K300 salt bridge in EcNhaA shifts the pKa of K300 down by 2 to 3 pKa units, which leads to a different view about the mechanism than proposed here.

I feel the authors should be clear about the limitations of their approach and acknowledge that dynamic effects could change the proposed picture.

- p9, L 210: PROPKA calculations on CHX17 and KefB (see also Fig S4).

- "estimate the pKa of the conserved lysine at pH 7.0" does not make sense. The pKa is independent of pH and only depends on the environment of the residue. Please correct.

- Can you provide a sense for the sensitivity of the pKa estimates, e.g. by running it on an ensemble of models or when applied to models based on different templates?

- p10, L 236 Ion selectivity hypothesis

The hypothesis about the location and nature of the ion selectivity "filter" is interesting.

- How do the authors envisage would this selectivity manifest itself, e.g., in binding to transport site or in the transport step itself?

- How do their results compare to the hypothesis of [R. Alhadeff, A. Ganoth, M. Krugliak, and I. T. Arkin. Promiscuous binding in a selective protein: the bacterial Na⁺/H⁺ antiporter. PLoS One, 6(10):e25182, 2011] who suggested that in EcNhaA, binding itself is not selective?

- p10, L252 D136N mutant rescue

- Interesting and encouraging results.

- Please also show the experimental data for the inactive D163N mutant in the SI.

- I'd really like to see better and direct evidence for the electroneutrality of the triple mutant as this would make this result much more impactful and useful, especially as the authors stated that it was designed to be electroneutral and on p14 L344 they interpret their results as "thus confirming the original goal of the single mutant design."

- p14, L353 "Our model for electrogenicity ..."

The paragraph talks about "masking charges". I don't understand what this is supposed to mean in terms of the actual physics and chemistry. Is this an argument about stability (and free energies) or strength of interactions? I would appreciate a more precise formulation of the authors' argument, at least to a degree that one could calculate some actual numbers using computational approaches.

- p15, ~L 378 "Collectively, our results are more consistent with the traditional antiporter mechanism"

- In Ref 24 an electroneutral mutant of TtNapA was described (K305Q). Wouldn't the authors' model suggest that TtNapA K305Q ought to be electrogenic as it (1) retains DD and (2) could still hydrogen bond between Q[6] - and K[8] (similar to what the authors

suggest for the 6-8 H-bond in AtCHX17 and PaKefB)?

- See also notes on Fig 4 below.

- HsNHA2 model

- Why was the HsNHA2 model based on MjNhaP1 and not on TtNapA? What is the sequence identity between HsNAH2 and TtNapA when computed in the same way as for MjNhaP1?

MjNhaP1 is a CPA1 in NhaP-III (Fig 1) and more distant from HsNHA2 than TtNapA (CPA2 NGC Fig 2). Just going by sequence identity (which is on the border line for homology models of membrane proteins) seems to ignore the authors' own findings on the evolutionary relationships.

(Given that an electroneutral transporter such as MjNhaP1 was chosen as template also makes it less surprising that HsNHA2 shows structural characteristics of an electroneutral transporter.)

- Does the model differ in key aspects when based on TtNapA? How robust are the conclusions against using different templates?

- TtNapA has outward facing and inward facing structures. The authors should build models in both conformations and show that the R432 - D278 salt bridge exists in both conformations --- at least this seems to be the conclusion to be drawn from their proposed mechanism.

- What is the predicted pKa of D[6] (and other key residues) in the model (and in inward- and outward facing conformations)?

How does this compare to the prediction that it should always be protonated (p15 L 385)?

- "Consistently, HsNHA2 did not become electrogenic even upon mutating the arginine at position 8 to lysine [Ref 24]" --- more precisely (see Table 1 in Ref 24), only very poor or no transport could be measured. I don't see how this supports the authors' point (especially as many mutants lose function for reasons that are typically not well understood).

- Fig 4, caption:

The statements about which residues change protonation states are speculative and not based on good evidence. The authors put forward a hypothesis but not an explanation based on evidence. Estimating pKas and changes of protonation states is very challenging (experimentally and computationally) and the discussion of Fig 4

(caption and text) uses a lot of "chemical intuition" to suggest changes in protonation states. It should be made clearer that these are *proposed* models and not confirmed mechanisms.

To expand on my point of the limitations of static structures and using chemical intuition: As the authors are probably aware, interactions with cations can induce localized changes that change pKas of residues substantially and such events do not seem to be taken into account in the authors interpretation of sequence data mapped on representative static X-ray structures.

A clear example where this situation is shown computationally is [Y. Huang et al. Mechanism of pH-dependent activation of the sodium-proton antiporter NhaA. *Nature Communications*, 7:12940, 10 2016] where the local pKas are computed with the currently best pKa method (constant pH MD simulations). The paper shows how the breaking of the D163-K300 salt bridge in EcNhaA shifts the pKa of K300 down by 2 to 3 pKa units.

Supplementary information:

Fig S3b: Shouldn't NspA/GerN be NapA/GerN (compare the caption)?

Fig S5: inset label _D164N should be _D163N

Fig S8 and S9 should be included into SI and not hosted on an external server in order to guarantee long-time availability.

SI Text: Expanded Results  Expanded (?) Results

Reviewer #3:

Remarks to the Author:

This manuscript attempts a large-scale phylogenetic analysis to identify the major families of cation/proton antiporters and then determine the residues in these proteins that are responsible for ion selectivity and electrogenicity. They identify 8 residues that they argue are important for determining these features of the proteins and discuss how the main split in the tree - between CPA1 and CPA2 doesn't exactly correlate with differences in these properties.

I will comment only on the phylogenetic aspects of the study. The authors do an extensive database search to collect ~15,000 homologs of these proteins. They then winnow down the number of representatives using a 70% identity threshold for CD-HIT as well as removing proteins missing key structural features yielding a set of ~7500. After some analyses using Fasttree2 and UPGMA, they used rogue-taxon identification in RogueNaRok to remove taxa in the tree that moved around in the tree a lot during the bootstrap analyses. The final set of sequences for the phylogenetic analyses had ~6400 representatives.

There are a number of problems with the foregoing approach. First, the authors are trying to do a rigorous and comprehensive analysis to improve on previous analyses. They do manage to gather a LOT of sequences. However, the problem that this number of sequences presents is that the

quality of the phylogenetic analyses they perform is seriously compromised. Truly rigorous phylogenetic analyses require the use of methods such as RAXML or IQ-TREE that do in-depth tree searching. Fasttree2 is a rapid approximation. UPGMA is a method that rarely if ever ever used by phylogeneticists because it is well known to be subject to systematic error when rates of evolution are not exactly clock-like amongst sequences (and such rates are NEVER clock-like). Consequently, the analyses reported are really sub-standard and not really acceptable. So what to do? Here are some suggestions:

- 1) the authors should take the trees they have estimated with Fasttree and, from each strongly supported clade, subselect the most divergent sequences. Do this in a way so that they can get the number of sequences in the analysis down to <500 or so. Then use better phylogenetic methods and models to analyze this abbreviated data set. Report the results of those analyses -- do they recover the same clades as the approximate methods with the full data set?
- 2) For the above analyses, the authors should try to consider more sophisticated phylogenetic models. For example they could try the LG4X model in RAXML and IQ-TREE. Even better would be if they tried the C10 or C20 mixture model (with the gamma distributions for rates across sites) in IQ-TREE -- these model capture site-specific features of proteins and therefore tend to better fit the data. If possible, the authors could even try model selection using IQ-TREE (including the mixtures). The general idea is to make sure that this sub-selected data set analyzed with better methods and models recovers the SAME main clades as the overall approximate methods did. This will substantially bolster the authors conclusions.
- 3) The UPGMA analysis should not even be discussed. Nobody will trust this.
- 4) The authors need to be more explicit about how they did analyses. For example, after alignment with the HMM, did they trim the ambiguously aligned regions with a program like TrimAL or BMGE or Zorro? IF not, they probably should. Secondly, they need to be specific about how the RogueNaRok analyses were conducted. What parameter settings were used? How were thresholds determined for deletion of unstable taxa?
- 5) The authors should always indicate how many aligned sites were analyzed with the various phylogenetic programs.

Regarding the Consurf analyses, I think the authors have done an ok job of describing them. They should be clear on the settings of all parameters in their analyses.

In summary, I think that if the above analyses are done and the original findings are robust to such analyses, then I think the manuscript will be much stronger.

Reviewer #1 (Remarks to the Author):

Major Comments:

(1) This is a very informative and useful study: a) to examine over 6 thousand sequences; b) to determine the molecular bases for separating CPA1 and CPA2, and c) to predict which residues mediate electrogenic exchange or not based on residues in the active motif and other expts. This is a very useful approach that is applicable to other transport proteins. As one can predict transport mode of poorly studied proteins based on active motifs of characterized homologs.

For transporters with high homology to EcNhaA, the conclusion would likely apply. Most prokaryote CPA2 have a DD motif whereas CPA1s have a –ND-. However, eukaryote CPA2s are mostly found in plants and some in fungi (Chanroj et al. 2012). CPA2 are rare or not found in most metazoan. HsNha2 is an exception.

(2) So it is questionable whether this blanket statement applies to other CPA2 which do not have a DD in TM5? reasons are:

a) Most eukaryote CPA2 do not have a DD motif. Only –ND- or –XD- with a few exceptions: e.g. HsNha2 (R (8)), or Dicdi-Nhe3 (K(8)) have a –DD- motif.

If Masrati et al.'s proposal is right, then Dicdi-Nhe3 should be electrogenic, and HsNha2 could be mutated to become an electrogenic antiporter.

b) There is considerable variability in residues at the 6 position: –DD- (6,7).

I.e. –ND-, –QD-, –TD-, –SD-. Within several plant CHX families, variation suggests this residue is not critical for separating CPA2 from CPA1. Other residues likely work together with the X(6).

Reply (comments 1 & 2)

We thank the reviewer for the positive feedback. Indeed, position 6 of the CPA motif emerged as relatively variable. As the reviewer points out this is very prominent in plants CHX transporters. In accordance with the reviewer's assertion that it is likely that other residues work together with

X₆, our model for electrogenicity requires two acidic residues in positions 6 and 7 of the motif and a basic residue in position 8. We revised the text to make this point clearer (see p. 14-15, L351-367).

It is also true that our hypothesis concerning electrogenicity is based only on prokaryotic antiporters, which have a D₆D₇ motif on TM5, as these are the only characterized electrogenic CPAs so far. It is thus not unlikely to propose that a different mechanism might exist in eukaryotic members of the CPA2 sub-family. We thus revised the text accordingly (see p. 15, L381-384).

(3) The most conserved residue among CPA2 is a K, Lysine, in TM10. Could K (8) form a salt bridge with another acidic residue (other than D (6))?

Check radius of 4Å around.

Reply

We thank the reviewer for this suggestion. Both in EcNhaA and TtNapA, the K (8) residue was crystalized in both inward- and outward-facing conformations. Yet, we could not identify any acidic residue that could potentially form a salt bridge with K₈ when scanning a radius of 4Å, other than D₆ (D163 in EcNhaA and D156 in TtNapA). We further analyzed the amino acids distribution in the multiple sequence alignment for all the positions within a radius of 4Å from K₈ in both EcNhaA and TtNapA, taking into consideration that these two transporters might not represent the rule and are rather the exception. Nevertheless, this analysis showed that in most cases acidic residues could be found in low frequencies (less than 0.6% in these positions). The only exceptions were two positions on TM3 where the frequency of aspartate and glutamate was between 4% to 7%. However, the sequences presenting these sequential characteristics are mammalian NHA-like transporters and related prokaryotic homologues that we discuss in the text and were the basis for our triple mutant's design (see p. 15-16, L385-492).

(4) Additional core residues may exist in the TM11- unwound region. Authors do not consider/discuss residues here, though K383 was shown to affect activity in the CPA2-type CHX17, as monitored by growth of yeast mutants (Czerny DD. et al. 2016. BBA). A basic residue

is conserved here as a K or R in several CPAs, e.g. TtNapA. It is likely a basic residue has some function?

(5) There is an additional conserved acidic residue near the TM11-unwound region of CPA2, several prokaryote CPA2 show a glutamate, though an acidic residue is not seen in EcNhaA. Is it possible, EcNhaA may not be representative of some CPA2 with respect to residues critical for electrogenicity?

Reply (comments 4 & 5)

We agree with the reviewer that there are additional conserved residues in CPA members that are not being address in this manuscript. As stated in the manuscript, our goal was to find a minimal set of amino acids that will enable to distinguish between CPA1s and CPA2s and between the two main CPA phenotypes (electrogenicity and ion selectivity). Future studies, though, might lead to the expansion of the motif beyond the eight positions introduced in this paper. As explained in the last two paragraphs in Discussion, we intend to try and characterize at least some of these conserved positions in future works.

As for the two titratable residues pointed out by the reviewer, indeed, our conservation analysis identified these two positions as conserved. We acknowledge the fact that due to their location in the transporter's core as part of the TM4-TM11 assembly, these residues should be discussed in the text regardless of the fact that they were not included in the motif. We revised the manuscript accordingly and discuss the equivalents of these two residues in TtNapA in the Methods section where we explain how the motif was defined (see Methods p. 24-25, L611-634). Specifically, the position corresponding to K383 in AtCHX17 (R331 in TtNapA) is more conserved in CPA1 members compare to CPA2s. As much as 90% of CPA1s are characterized by a basic residue in this position, compared to roughly 30% of CPA2s. Thus, the distinction between CPA1s and CPA2s cannot be based on this position. As the reviewer pointed out, in CPA2s, such basic residue could be found in NapA-like transporters. As NapA is electrogenic and CPA1 are electroneutral, this position is also not a good marker for electrogenicity. Finally, this basic residue could be found in both Na⁺- and K⁺-selective CPA1s and thus does not confer ion selectivity. That being said, this conserved position could still be structurally important, which will explain the loss of function resulting from its substitution with alanine in AtCHX17.

As for the glutamate residue on TM-11 that is absent from EcNhaA (E333 in TtNapA). Although this position is highly conserved in CPA2s, it is unlikely to mediate proton transport as it cannot explain the existing experimental data, as follows. Drew and colleagues demonstrated that the replacement of K305 with glutamine in TtNapA resulted in an electroneutral transport. While this could, in theory, imply a direct involvement of K305 in electrogenic transport, as the proton carrier, it may also be the result of an indirect action of K305. That is, K305 may act by modulating the dielectric environment of an adjacent titratable residue, which is the component that determines electrogenicity. Our experimental data supports the indirect involvement of K305. As K305 is $\sim 12\text{\AA}$ apart from E333 in both the inward- and outward-facing conformations of TtNapA, we find it unlikely that the latter could participate in proton transport either directly or indirectly. Located close to the ion binding site, E333 could potentially participate in ion coordination, however, as it can be found in both Na⁺- and K⁺-selective CPA2s it does not confer ion selectivity and was not included in the motif.

(6) Proposed Experiment: The idea would be more convincing with additional experiment. If the hypothesis of Masrati et al is correct, then HsNha2 (electroneutral) should be convertible to electrogenic antiport by converting R (position 8) to a K. HsNha2 has a DD motif (TM5), but an R (8) instead of K. Line 391. Authors propose this expt in the Discussion. Inclusion of this experiment would be critical. Alternatively, show that transport of a plant CHX (CPA2) is electroneutral.

Reply

We agree with the reviewer that further experimental analysis of putative electroneutral/electrogenic eukaryotic CPAs is important for extending our conclusions to eukaryotic transporters. We are currently unable to carry out these experiments because the Padan lab, who conducted all the reported experiments, focuses on EcNhaA only. But we seek new collaborations to conduct these experiments in the future. However, we strongly believe that our computational analysis and existing experiments describe best the mechanism for electrogenicity in prokaryotic CPAs, and we emphasize in the manuscript that future experiments will be required to explore its validity to eukaryotes (as mentioned above in reply to comments 1 and 2, see p. 15, L381-384).

Minor comments:

(a) Fig. 2. Does the subclade CHX refer to mostly plant proteins? If so, is the motif for D6-D7 in error? Most plant CHX genes encode –X6-D7-. If authors refer to prokaryotes only, then indicate.

This reviewer is not aware of –DD- which is probably very rare in eukaryote CHX TM5. Please revise.

Reply

Indeed, most plants CHX genes and other eukaryotic CHX-like transporters encode X₆D₇ motif (in particular, roughly 50% present an ND motif), while the D₆D₇ motif is a characteristic of prokaryotic genes that are part of this clade. In light of the reviewer's comment we revised figure 2 to reflect this point.

(b) Fig. S6. Method for measuring transport activity. Typos probably in reaction mixture regarding milliMolar versus microMolar. E.g. protein at 50-100 mg? Acridine Orange at 0.5 mM.??

Reply

Typos were corrected. Figure S6 was renumbered in the revised text and is now figure S4.

(c) Line 144. Plant CHXwere. 'never shown to be electrogenic'. This statement is misleading. So far there has been no direct transport assays of any plant CHXs. So the possibility has yet to be tested. The statement might be revised to: 'Whether plant CHXs are electrogenic or not has yet to be established.'

Reply

We agree. The text revised accordingly (see p. 6 L148-149).

(d) Other differences between CPA1 and CPA2: Would authors discuss if the basis for different pH set points can be identified in the structure? There is E78 in TM2 EcNhaA which is conserved in many CPA2, but not in CPA1. If this is a pH sensor, how would it affect electrogenic versus electroneutral transport?

Reply

Generally speaking, our phylogenetic analysis can be used to predict locations within the sequence of the transporters, which may be responsible for pH sensing. While outside the scope of the manuscript, this is certainly an interesting question, and we will try to investigate it in future studies.

As for E78 in EcNhaA, indeed, this residue is highly conserved in CPA2s (75% of the representatives) and could also be found in roughly 12% of CPA1s. However, the equivalent glutamate in TtNapA was shown to have no effect on the pH dependency of the transporter (Furrer, E. M., et al. (2007). "Functional characterization of a NapA Na⁺/H⁺ antiporter from *Thermus thermophilus*." *FEBS letters* **581**(3): 572-578.).

Reviewer #2 (Remarks to the Author):

Major Comments

(1) p3, L6: stoichiometry *could* be different from 1:1 or 2:1

Reply

To the best of our knowledge, a stoichiometry different from 1:1 or 2:1 was reported for the NhaB transporter from *E. coli* where a 2Na⁺ ions are exchanged for 3H⁺ (Pinner, Elhanan, Etana Padan, and Shimon Schuldiner. "Kinetic properties of NhaB, a Na⁺/H⁺ antiporter from *Escherichia coli*." *Journal of Biological Chemistry* **269**, no. 42 (1994): 26274-26279). However, despite the misleading name and the frequent references to NhaB as part of the CPA superfamily, the sensitive homology detection and structure prediction tool HHpred predicts that NhaB have a fold different from the NhaA-fold. The best scoring structural-homologue of EcNhaB in the PDB is the divalent anion/Na⁺ symporter from *Vibrio cholera* (PDB id 5ULD), where, indeed, two Na⁺ binding sites were

reported. EcNhaA, on the other hand, was not identified even as a remote structural-homologue. Our manuscript focuses only on those CPAs that share the NhaA-fold, as this structural similarity is crucial for our analysis. Thus, EcNhaB and similar transporters were not included in this work. We revised the text such that our definition of the CPA superfamily will reflect that (see p. 4, L102-103)

(2) p4, L86 abstract: "*CPA1/CPA2 division only partially correlates with electrogenicity*"

The authors state that it has not been clear to many in the field of CPA transporters that the CPA1/CPA2 families do not strictly correlate with electroneutral or electrogenic transport. The paper makes an important contribution driving home this point.

If find it less convincing to specifically single out the paper by Brett et al 2005 (Ref 20) as having put forward this view. Although there is a sentence in the paper where the authors suggest a CPA2 transporter should be looked for electrogenic transport in crab gills, they also state that, for example, in fungi, no electrogenic NHAs (CPA2) are known. So on balance, my reading of Brett 2005 is more in line with what the authors also propose (notwithstanding the fact that the present paper now places the fungal NHAs closer to the eukaryotic NHEs in the CPA1 clade than the animal NHAs in the CPA2 clade).

Reply

We thank the reviewer for this comment. We updated the text and references to reflect this point (see p4, L86 references 6,8,22,23).

(3) p7, L158: "*Because the tree can reproduce the CPA1/CPA2 division...*"

It is not clear from the main text that the fungal NHAs were re-assigned from CPA2 to CPA1 (Ref 20 considered then part of the CPA2 family). The supplementary text discusses this in more detail but is easy to overlook. Perhaps make clearer that the reassignment of fungal NHAs to CPA1 a new result of this paper.

It is therefore at least technically not true that "the tree can reproduce the CPA1/CPA2 division" because the authors change the CPA1/CPA2 division. Make clearer which results confirm existing knowledge and which ones update or challenge it.

Reply

We thank the reviewer for highlighting this issue. To address this, the following text was added to the CPA2 clades description: "This clade was previously assigned to the CPA2 sub-tree²¹. However, based on the conserved sequential features discussed below, the assignment of fungal NHAs to the CPA1 sub-tree is more appropriate." (p. 5-6, L129-132). Following this, as we describe the sequential features that characterize CPA1s we state that: "This feature is also found in fungal NHAs, which were previously classified as CPA2s²¹. Based on our work, these transporters should be assigned to the CPA1 sub-tree." (p. 8, L207). As for the phylogenetic tree's ability to reproduce the CPA1/CPA2 division we revised as follow: "Because the tree can reproduce the CPA1/CPA2 division, with the exception of the fungal NHA clade..." (p. 7, L162-163).

(4) p8, L 206 "position 8 is populated by lysine (K300), that salt bridges with aspartate at position 6 (D163)"

The Lee et al (2014) [Ref 6] EcNhaA crystal structures show the mentioned salt bridge. However, there are inherent limitations to trying to derive a mechanism from static structures and supposing that the salt bridge always exists might not do justice to the actual molecular mechanism.

The "salt bridge breaking model" [Ref 6] proposed (based on MD simulations) that interactions with cations can induce localized changes that change the pKa of K300 substantially. These results were computationally validated recently [Y. Huang at al. Mechanism of pH-dependent activation of the sodium-proton antiporter NhaA. Nature Communications, 7:12940, 10 2016] where the local pKas were computed with one of the currently best pKa method (constant pH

MD simulations). The paper shows how the breaking of the D163-K300 salt bridge in EcNhaA shifts the pKa of K300 down by 2 to 3 pKa units, which leads to a different view about the mechanism than proposed here.

I feel the authors should be clear about the limitations of their approach and acknowledge that dynamic effects could change the proposed picture.

Reply

We appreciate the reviewer's valuable insight and we fully agree that dynamic effects could have profound implications on the mechanism. We also acknowledge the fact that our model could be one of several possible models, and that there is always the possibility that not all CPAs share the same transport mechanism, especially in light of the great diversity among the members of this family. In light of this, we revised the results and discussion sections to better reflect the limitations of our approach. Specifically, we now emphasize that our analysis is mostly in the sequence level and that any further inferences are putative and based on the available CPA structures and experimental data (see for example p. 8-9, L198-209; p. 13 L313-319; p. 14-15, L351-367; p. 15, L381-384).

(5) p9, L 210: PROPKA calculations on CHX17 and KefB (see also Fig S4). "estimate the pKa of the conserved lysine at pH 7.0" does not make sense. The pKa is independent of pH and only depends on the environment of the residue. Please correct.

(6) Can you provide a sense for the sensitivity of the pKa estimates, e.g. by running it on an ensemble of models or when applied to models based on different templates?

Reply (comments 5 & 6)

We thank the reviewer for this important comment. Following the reviewer's request, we tested hundreds of different models generated using MODELLER and pairwise alignments deduced from the MSA that was used to reconstruct the phylogenetic tree. These, included ensembles of models that were based on the same template as well as models that were based on different

templates. These templates included the crystal structures of TtNapA (PDB ids 5BZ and 5BZ3), MjNhaP1 (PDB id 4CZB) and PaNhaP (PDB ids 4CZ8 and 4CZ9). Models based on the structure of EcNhaA were not assessed as it emerged as an inferior template for modelling CHX17 and KefB. As it turned out, the PROPKA calculations were very sensitive even to the small differences between models that were based on the same template. Given these results we decided to exclude the PROPKA calculations from the manuscript. Nevertheless, even without these calculations, we still believe our proposed model for electrogenicity best explains both old and new experimental results as discussed in the text (see p. 14-15, L351-367).

(7) p10, L 236 *Ion selectivity hypothesis*

The hypothesis about the location and nature of the ion selectivity "filter" is interesting.

How do the authors envisage would this selectivity manifest itself, e.g., in binding to transport site or in the transport step itself?

How do their results compare to the hypothesis of [R. Alhadeff, A. Ganoth, M. Krugliak, and I. T. Arkin. Promiscuous binding in a selective protein: the bacterial Na⁺/H⁺ antiporter. PLoS One, 6(10):e25182, 2011] who suggested that in EcNhaA, binding itself is not selective?

Reply

We thank the reviewer for the positive feedback. The hypothesis presented in Alhadeff et al. that both potassium and sodium bind to the protein with the same energy, do not contradict our findings that implicated an extra polar amino acid in the selectivity of K⁺-specific CPAs. This is because the calculations of Alhadeff et al. were carried out on a single state of the transporter (inward-open) and sampled conformations around this state. The entire transport process involves other states that could potentially be responsible for the selectivity (for example, semi-occluded and occluded states). That is, differences in the binding energies of the different ions, in any of the other states, may give rise to kinetic barriers that would manifest as different transport rates. Since our model is based on sequence features alone, the change in the binding energy of the potassium ion due to the added polar amino acid, and resulting change in transport rate, could

occur in any of the states occupied by the protein during the transport process. This also includes the states that were not addressed in the simulations of Alhadeff et al.

Generally speaking, in channels, selectivity is commonly explained by the ability of one state of the protein (open state) to compensate for the loss of hydration by providing coordinating groups in a way that favors one type of ions. In the case of transporters, however, selectivity should result from the ability of only one ion to support the structural and dynamic changes in the protein that accompanied productive transport. This support is applied through non-covalent interactions between the ion and the transporter, which create in some of the stages energetically stable complexes and in other stages unstable complexes that induce conformational changes in the protein. Thus, fully understanding the mechanism of selectivity in any given transporter require a rigorous analysis of the structural and energetic aspects of the protein-ion complex throughout the entire transport trajectory. Unfortunately, space limitations left us no alternatives but to discuss this important issue in the Suppl (section "Expanded discussion" p. 25-26, L456-484).

(8) p10, L252 D136N mutant rescue

Interesting and encouraging results. Please also show the experimental data for the inactive D163N mutant in the SI.

I'd really like to see better and direct evidence for the electroneutrality of the triple mutant as this would make this result much more impactful and useful, especially as the authors stated that it was designed to be electroneutral and on p14 L344 they interpret their results as "thus confirming the original goal of the single mutant design."

Reply

We thank the reviewer for the positive feedback and for this important suggestion. Following the reviewer's comment, we added the experimental data for the inactive D163N mutant. It could be found in Fig, S3a, where we show that in inverted membrane vesicles expressing the D163N mutant there is no Li⁺-induced proton efflux.

Furthermore, to confirm the electroneutrality of the P108E-A160S-D163N triple-mutant, we tested the effect of valinomycin, a potassium ionophore, on the Na⁺-induced proton efflux through the transporter. Electrogenic CPAs produce a positive out $\Delta\Psi$ in everted membrane vesicles that limits the rate of the antiporter activity. By dissipating this charge, the presence of valinomycin/K⁺ accelerate the antiporter's rate. In contrast, this manipulation should not affect the rate of an electroneutral antiporter. The results summarized in figure 6 show that the presence of valinomycin/K⁺ has a drastic effect on the rate of the WT but hardly any effect on the rate of the triple mutant. Taken together, our results confirmed our prediction that the triple mutant is electroneutral (see p. 12, L. 286-300 and Fig. 6).

(9) p14, L353 "Our model for electrogenicity ..."

The paragraph talks about "masking charges". I don't understand what this is supposed to mean in terms of the actual physics and chemistry. Is this an argument about stability (and free energies) or strength of interactions? I would appreciate a more precise formulation of the authors' argument, at least to a degree that one could calculate some actual numbers using computational approaches.

Reply

We agree that this paragraph was a bit hard to interpret. We were trying to discuss the physical effect of surrounding charges and dielectric environment on the apparent pKa of titratable residues in the transporter's binding site. However, upon further consideration we decided to rewrite this paragraph, in light of our decision to discard the PROPKA calculations (see replay to comments 5-6) and the fact that our analysis was mainly based on sequence-related data (see p. 14-15, L351-367).

(10) p15, ~L 378 "Collectively, our results are more consistent with the traditional antiporter mechanism"

In Ref 24 an electroneutral mutant of TtNapA was described (K305Q). Wouldn't the authors' model suggest that TtNapA K305Q ought to be electrogenic as it (1) retains DD and (2) could still hydrogen bond between Q[6] - and K[8] (similar to what the authors suggest for the 6-8 H-bond in AtCHX17 and PaKefB)?

See also notes on Fig 4 below.

Reply

Following the reviewers' comments, we realized that our description of the model for electrogenicity may not have been clear enough. We thus completely revised the relevant part in "Discussion" (see p. 14-15, L 351-367). Simply put, we hypothesize that both an acidic residue at positions 6 and 7 of the CPA motif and a basic residue at position 8 determine electrogenicity. Based on our experimental data, we further suggest that the acidic residues at position 6 and 7 are the protons carriers. As for the basic residue at position 8, it might affect the dielectric environment, which in turn modulates the pKa of position 6 and allow for protonation and deprotonation, resulting in proton transport. Thus, the K305Q TtNapA mutant is expected to be electroneutral according to our model, as it lacks the basic residue at position 8. As for AtCHX17 and PaKefB, both lack the acidic residue at position 6 that we show experimentally is the most likely proton carrier, and thus are also electroneutral.

(11) HsNHA2 model

- Why was the HsNHA2 model based on MjNhaP1 and not on TtNapA? What is the sequence identity between HsNHA2 and TtNapA when computed in the same way as for MjNhaP1?

MjNhaP1 is a CPA1 in NhaP-III (Fig 1) and more distant from HsNHA2 than TtNapA (CPA2 NGC Fig 2). Just going by sequence identity (which is on the border line for homology models of membrane proteins) seems to ignore the authors' own findings on the evolutionary relationships.

(Given that an electroneutral transporter such as MjNhaP1 was chosen as template also makes it less surprising that HsNHA2 shows structural characteristics of an electroneutral transporter.)

- Does the model differ in key aspects when based on TtNapA? How robust are the conclusions against using different templates?

- TtNapA has outward facing and inward facing structures. The authors should build models in both conformations and show that the R432 - D278 salt bridge exists in both conformations --- at least this seems to be the conclusion to be drawn from their proposed mechanism.

- What is the predicted pKa of D[6] (and other key residues) in the model (and in inward- and outward facing conformations)?

How does this compare to the prediction that it should always be protonated (p15 L 385)?

Reply

Our HsNHA2 model was based on the archaeal CPA1 member MjNhaP1 as it emerged as the best template available in the PDB, based on the accurate homology detection tool HHpred. MjNhaP1 presents a 19% sequence identity to HsNHA2, compared to 16% sequence identity between TtNapA and HsNHA2. However, MjNhaP1 and TtNapA present a remarkable structural homology (TM-score of 0.85 and RMSD of 2.98Å calculated for 4CZB and 5BZ2 by TM-align). This similarity is even greater than the similarity between TtNapA and EcNhaA (TM-score of 0.69 and RMSD of 4.06 Å calculated for 4AU5 and 5BZ2 by TM-align), both well-established CPA2 members. In this sense, both structures (MjNhaP1 and TtNapA) could have been used as suitable templates for modelling HsNHA2. As structure tends to be more conserved than sequence, it is plausible that some CPA1s and CPA2s will share similar structures regardless of their sequential differences that are reflected in the phylogeny. Thus, we do not believe that these results are in conflict with the phylogenetic analysis.

However, we agree with the reviewer that the choice in a CPA1 member for modeling might seem a bit odd to the general reader. We thus modeled HsNHA2 in both inward and outward facing conformations based on the structures of TtNapA (PDB ids 5BZ2 and 5BZ3, and

see Methods p. 20-21, L515-532). The conclusions drawn from the new models concerning the putative electrostatic interactions between E215, D278 and R432 are practically the same. E215 could, potentially, salt bridge with R432. To further test the sensitivity of our results we produced an ensemble of 100 models based on the inward facing 5BZ2 crystal structure, and another ensemble of 100 models based on the outward facing 5BZ3 crystal structure. These represent different side chain rotamers of E215 and R432. In 92% of the models based on 5BZ2 and 79% of the models based on 5BZ3, E215 and R432 are at a distance lower than 4Å and thus, potentially, could salt bridge. The hypothesis that these two positions could interact is also indirectly supported by our EcNhaA triple mutant. However, considering the fact that the PROPKA calculations emerged as highly sensitive even to small differences between models based on the same templates, we cannot commit to the protonation states of these three residues. Nevertheless, considering the work of Drew and colleagues that showed HsNHA2 to be electroneutral, and our new results demonstrating that D₆ is the second proton carrier, we believe that our hypothesis concerning the electroneutrality of HsNHA2 is at least feasible.

(12) "Consistently, HsNHA2 did not become electrogenic even upon mutating the arginine at position 8 to lysine [Ref 24]" --- more precisely (see Table 1 in Ref 24), only very poor or no transport could be measured. I don't see how this supports the authors' point (especially as many mutants loose function for reasons that are typically not well understood).

Reply

We agree with the reviewer's comment and removed this statement. The poor transport measured for the K305R mutant in TtNapA is irrelevant to the electrogenicity model. Together with the K300R mutant in EcNhaA that also showed reduced activity, it seems to reflect the natural preference to K over R at position 8 of the motif in CPA2s, observed in the phylogeny. We thus removed this sentence.

(13) Fig 4, caption:

The statements about which residues change protonation states are speculative and not based on good evidence. The authors put forward a hypothesis but not an explanation based on evidence.

*Estimating pKas and changes of protonation states is very challenging (experimentally and computationally) and the discussion of Fig 4 (caption and text) uses a lot of "chemical intuition" to suggest changes in protonation states. It should be made clearer that these are *proposed* models and not confirmed mechanisms.*

To expand on my point of the limitations of static structures and using chemical intuition: As the authors are probably aware, interactions with cations can induce localized changes that change pKas of residues substantially and such events do not seem to be taken into account in the authors interpretation of sequence data mapped on representative static X-ray structures.

A clear example where this situation is shown computationally is [Y. Huang at al. Mechanism of pH-dependent activation of the sodium-proton antiporter NhaA. Nature Communications, 7:12940, 10 2016] where the local pKas are computed with the currently best pKa method (constant pH MD simulations). The paper shows how the breaking of the D163-K300 salt bridge in EcNhaA shifts the pKa of K300 down by 2 to 3 pKa units.

Reply

We fully agree with the reviewer's comment. Indeed, predicting protonation states of titratable residues is a challenging task, especially considering the changes in the dielectric of the protein's core, resulting from dynamic processes such as conformational changes and ion binding. In light of this we revised the text so it will better reflect the fact that our data is mostly sequential. Specifically, we removed any references to Fig. 4 (now Fig. 7) from the results section, and revised the figure's legend, emphasizing that the proposed model is a hypothetical one. Finally, we did our best to avoid what the reviewer refers to as "chemical intuition" throughout the text and clearly separate the data from our interpretation of it. As such, we discarded PROPKA's calculation for the reasons described above and revised parts of the results (see for example p. 8-9, L198-209) and discussion sections (see for example p. 13, L313-319 and p. 14-15, L351-367).

Minor Comments:

Supplementary information:

(a) Fig S3b: Shouldn't NspA/GerN be NapA/GerN (compare the caption)?

Reply

Figure S3 (now Fig. S6) was updated and the mistake corrected.

(b) Fig S5: inset label _D164N should be _D163N

Reply

Fixed. D164N in Fig. S5 (now Fig. S4) was changes to D163N.

(c) Fig S8 and S9 should be included into SI and not hosted on an external server in order to guarantee long-time availability.

Reply

Fig S8 and S9 (now S10, S11 and S12) were revised and are now included in the supplementary information.

(d) SI Text: Expanded Results  Expanded (?) Results

Reply

Typo was corrected.

Response to Reviewer #3 (Remarks to the Author):

Major Comments:

(1) the authors should take the trees they have estimated with Fasttree and, from each strongly supported clade, subselect the most divergent sequences. Do this in a way so that they can get

the number of sequences in the analysis down to <500 or so. Then use better phylogenetic methods and models to analyze this abbreviated data set. Report the results of those analyses -- do they recover the same clades as the approximate methods with the full data set?

(2) For the above analyses, the authors should try to consider more sophisticated phylogenetic models. For example, they could try the LG4X model in RAXML and IQ-TREE. Even better would be if they tried the C10 or C20 mixture model (with the gamma distributions for rates across sites) in IQ-TREE -- these models capture site-specific features of proteins and therefore tend to better fit the data. If possible, the authors could even try model selection using IQ-TREE (including the mixtures). The general idea is to make sure that this sub-selected data set analyzed with better methods and models recovers the SAME main clades as the overall approximate methods did. This will substantially bolster the authors conclusions.

Reply (comments 1 & 2)

We thank the reviewer for the insightful suggestions. Following the reviewer's recommendations, we repeated the phylogenetic analysis applying a more sophisticated phylogenetic model and used a more rigorous phylogenetic analysis tool on our complete data set as well as on a sub-selection of the 500 most divergent sequences.

More specifically, in order to choose the substitution model that best fits the data, we applied IQ-TREE's model finder tool (including mixture models) to the full list of representative sequences we collected. We also took this opportunity to introduce sequences that were not included in the previous analysis as a result of the clustering procedure. IQ-TREE predicted the best model to be the LG substitution model with empirical amino acids frequencies calculated from the data and a FreeRate model with 10 categories instead of the standard Gamma distribution. First, an initial phylogenetic tree was built (7499 representatives) using IQ-TREE and the LG+F+R10 model, followed by bootstrap analysis with 100 replicas. Next, RogueNaRok was utilized to identify rogue taxa that affect the support values of the inferred tree. By default, RogueNaRok searches for detrimental taxon one by one. Nevertheless, there may be groups of several taxon that should be pruned together (dropsets) in order to observe improvement in the tree's support. RogueNaRok's "-s" parameter allows such option, however for our large dataset a search for dropsets larger than two taxa became computationally highly expensive. Thus, at first,

we invoked RogueNaRok with $-s$ equals 2, and removed the identified rogue taxa as they all had an impact on the bootstrap value of at least one of the main clades of the tree. A main clade was defined as the largest clade with a relatively high bootstrap value (60%) that is close to the tree's base. The tree's base was defined by its division to two main sub-trees (later shown to represent the CPA1 and CPA2 sub-families) that are consistently recovered regardless of the method and/or model used. In order to estimate the effect of dropsets larger than two on the support of these main clades, we color-coded each clade and visually detected dropsets that were also removed. Potentially, there were other dropsets that affect the support values within each of these clades, however, this is beyond the scope of this paper. The final phylogenetic tree consisted of 6537 taxa.

To test whether the new analysis reproduces the same main clades previously identified, we annotated each taxon in the new phylogenetic tree according to its assignment in our original FastTree analysis. Overall, the same main clades were retrieved, that is, taxa that were clustered to the same clade in the FastTree analysis were clustered together also in the IQ-TREE analysis, with few minor differences (see figure S2). Specifically, in the CPA2 sub-tree, the NGC clade observed in the FastTree analysis that included the NapA, GerN and CHX sub-clades was split into four independent clades. These include the CHX clade, the GerN clade and two NapA clades (see Fig. 2). As for the GerN clade, some of the sequences assigned to this branch of the tree in the FastTree analysis were clustered outside of it in the IQ-TREE analysis. However, the main feature of the GerN clade is the affiliation of its sequences to the firmicutes phylum. In accordance with that, all sequences that were exempt from this clade in the IQ-TREE analysis do not belong to the firmicutes phylum. Similarly, in the CPA1 sub-tree, the NhaP-II Na⁺-specific clade observed in the FastTree analysis was split in the IQ-TREE analysis into two smaller clades – archaeal and bacterial NhaP-II Na⁺-specific (see Fig. 1). To further assess our results, we performed an additional IQ-TREE analysis on the 500 most divergent sequences, detected using the phylogenetic diversity analysis tool PDA (Chernomor, O., et al. (2015). "Split diversity in constrained conservation prioritization using integer linear programming." *Methods in ecology and evolution* 6(1): 83-91). IQ-TREE was then utilized for tree reconstruction and bootstrap analysis. Overall, all three phylogenetic trees reproduced the same main CPA clades with minor differences (Figs. S2 and S7). The Methods section was revised accordingly (see p. 19, L468-470; p. 19-20, L481-513).

(3) The UPGMA analysis should not even be discussed. Nobody will trust this.

Reply

We agree. Following this comment, the UPGMA analysis was discarded.

4) The authors need to be more explicit about how they did analyses. For example, after alignment with the HMM, did they trim the ambiguously aligned regions with a program like TrimAL or BMGE or Zorro? IF not, they probably should. Secondly, they need to be specific about how the RogueNaRok analyses were conducted. What parameter settings were used? How were thresholds determined for deletion of unstable taxa?

Reply

We revised the Methods section accordingly (see p. 19, L468-470 and p. 19-20, L481-513).

As for trimming the multiple sequence alignment, we are aware that it is sometimes argued that removing gap-rich regions is expected to improve the phylogenetic analysis. However, when working with such a diverse family, where the sequence identity between different members can be as low as 10%, there is a greater risk that the trimming procedure will result in the loss of important phylogenetic information. Thus, in this case, the success of such approach will depend immensely in choosing the right trimming parameters, such as the cutoff for removing columns. However, in most cases these parameters are chosen arbitrarily. Moreover, a recent study by Dessimoz and colleagues (Tan, G., et al. (2015). "Current methods for automated filtering of multiple sequence alignments frequently worsen single-gene phylogenetic inference." *Systematic biology* **64**(5): 778-791) thoroughly examined a list of automated filtering tools of multiple sequence alignments with different parameters, including TrimAl, BMGE and zorro, and their effect on the quality of the phylogenetic analysis. Overall, their findings show that trimming the alignment results in no improvement of the phylogenetic analysis at best, and on average trimmed alignments actually produced worse phylogenetic trees. In light of these findings we decided against trimming the alignment. Concerning the RogueNaRok analysis, we now better describe the exact procedures (p. 19-20, L490-503; see also previous comment).

(5) The authors should always indicate how many aligned sites were analyzed with the various phylogenetic programs.

Reply

Done. This information was added to the methods section (see p. 20 L502-503).

(6) Regarding the Consurf analyses, I think the authors have done an ok job of describing them. They should be clear on the settings of all parameters in their analyses.

Reply

ConSurf's analysis description was revised to include any missing parameters (see p. 22 L545-549).

In summary, I think that if the above analyses are done and the original findings are robust to such analyses, then I think the manuscript will be much stronger.

Reply

We thank the reviewer for his insightful comments, which helped us to significantly improve our analyses.

Reviewers' Comments:

Reviewer #1:

Remarks to the Author:

The revised manuscript clarifies most points raised by the reviewers. I have only a few minor comments for the authors.

1. It is clear the CPA2 versus CPA1 classification based on sequence homology is a result of diversification and is not a reliable criterion to distinguish functional distinctions, like electroneutral or electrogenic transport. This is an important conclusion though the impact is softened.

Wording changes are suggested:

a) Introduction:

L 85. "These analyses divided CPAs into CPA1 and CPA2 which occasionally are 'argued' (or stated) to correlate (relate to) with' the phenotypical electroneutral/electrogenic partition."

L. 94. "partially correlates" suggest revise to "partially corresponds".

'Correlate' usually infers a quantitative relationship which is not the case here.

L 87. Common dogma: 'Dogma' is too strong a word for this perspective. Consider "statement", or idea

2. The strategy to generate a triple EcNhaA mutant P108E, A160S, D163N which restores electroneutral Na/H activity to the single D163N mutant is very interesting, and is quite convincing with the valinomycin expt. It is not clear if the Km and Vmax of Na transport is altered in the mutant.

The electroneutral exchange would lend support to the idea that DD is important for electrogenic transport. Yet this is probably not the end of the story, as other studies (Huang et al 2016; and Uzdavinys et al. 2017) suggest a critical role for K300 as well.

As I mentioned in the first review, the idea would be more convincing if HsNha2 (electroneutral) is convertible to electrogenic antiporter by converting R (position 8) to a K. Future studies to address this are considered.

3. Supplemental information is very informative and useful for the community:

I have a few suggestions for minor revision.

a) Supp. L. 406. Plant AtKEA3 is not localized to the Golgi as stated in the supp info.

Please cite: Plastidial transporters KEA1, -2, and -3 are essential for chloroplast osmoregulation, integrity, and pH regulation in Arabidopsis.

Kunz HH, Gierth M, Herdean A, Satoh-Cruz M, Kramer DM, Spetea C, Schroeder JI.

Proc Natl Acad Sci U S A. 2014 May 20; 111(20):7480-5. doi: 10.1073/pnas.1323899111

b) Supp Materials, line 350-351. References cited missed one paper that is the most thorough functional characterization of plant AtCHX17, CHX18 and CHX19 in yeast, and expression in E coli. To date, this is the only study showing AtCHX17 can complement yeast mutant kha1p functionally.

Please cite:

Plant-specific cation/H⁺ exchanger 17 and its homologs are endomembrane K⁺ transporters with roles in protein sorting.

Chanroj S, Lu Y, Padmanaban S, Nanatani K, Uozumi N, Rao R, Sze H.

J Biol Chem. 2011 Sep 30; 286(39):33931-41. doi: 10.1074/jbc.M111.252650.

Manuscript section: Typo in Fig. 2

Fig. 2. CPA2 tree. Label at right side: is it 'GerN', instead of GreN?

Overall, this ms makes an important contribution as EcNhaA has been the leading model of like transporters which are found in all domains of life. Yet the mode of transport of CPAs and their biological roles in eukaryotes are largely unknown. The paper highlights key residues and predicts transport mode of most sequenced CPAs on earth. Moreover, experiments showed an electrogenic antiporter can be changed to an electroneutral one via mutation of a few strategic residues.

Reviewer #2:

Remarks to the Author:

The authors addressed all my comments. This is now a more nuanced and more solid paper. I'm sure it will generate interest in the field.

Reviewer #3:

Remarks to the Author:

My original comments on the manuscript were focused on the phylogenetic analyses. The authors have substantially clarified what they did in the Methods section and have conducted the requested analyses. I'm satisfied with the revised version of the manuscript.

Reviewer #1 (Remarks to the Author):

The revised manuscript clarifies most points raised by the reviewers. I have only a few minor comments for the authors.

1. It is clear the CPA2 versus CPA1 classification based on sequence homology is a result of diversification and is not a reliable criterion to distinguish functional distinctions, like electroneutral or electrogenic transport. This is an important conclusion though the impact is softened.

Wording changes are suggested:

a) Introduction:

L 85. "These analyses divided CPAs into CPA1 and CPA2 which occasionally are 'argued' (or stated) to correlate (relate to) with' the phenotypical electroneutral/electrogenic partition."

L. 94. "partially correlates" suggest revise to "partially corresponds".

'Correlate' usually infers a quantitative relationship which is not the case here.

L 87. Common dogma: 'Dogma' is too strong a word for this perspective. Consider "statement", or idea

Reply

We revised the Introduction accordingly.

2. The strategy to generate a triple EcNhaA mutant P108E, A160S, D163N which restores electroneutral Na/H activity to the single D163N mutant is very interesting, and is quite convincing with the valinomycin expt. It is not clear if the K_m and V_{max} of Na transport is altered in the mutant.

Reply

The analysis of Na⁺/H⁺ antiporter activity was conducted in everted membrane vesicles using acridine orange as a ΔpH probe. In this assay, a good estimate of the apparent *K_m* is obtained from the cation concentration that yields 50% activity and this concentration hardly changed for the triple mutant. Figure S4 shows that the *V_{max}* has changed. Most importantly, as opposed to the WT of which rate changes dramatically in the presence of valinomycin/K, the rate of the mutant does not change (see, Methods p. 28, L774-779).

The electroneutral exchange would lend support to the idea that DD is important for electrogenic transport. Yet this is probably not the end of the story, as other studies (Huang et al 2016; and Uzdavynys et al. 2017) suggest a critical role for K300 as well.

As I mentioned in the first review, the idea would be more convincing if HsNha2 (electroneutral) is convertible to electrogenic antiporter by converting R (position 8) to a K. Future studies to address this are considered.

Reply

We are seeking collaborations to this end. Hopefully for the next paper.

3. Supplemental information is very informative and useful for the community:

I have a few suggestions for minor revision.

a) Supp. L. 406. Plant AtKEA3 is not localized to the Golgi as stated in the supp info.

Please cite: Plastidial transporters KEA1, -2, and -3 are essential for chloroplast osmoregulation, integrity, and pH regulation in Arabidopsis.

Kunz HH, Gierth M, Herdean A, Satoh-Cruz M, Kramer DM, Spetea C, Schroeder JI.

Proc Natl Acad Sci U S A. 2014 May 20;111(20):7480-5. doi: 10.1073/pnas.1323899111

Reply

We changed the text and citations accordingly (see Supplementary Information p. 24, head of page).

b) Supp Materials, line 350-351. References cited missed one paper that is the most thorough functional characterization of plant AtCHX17, CHX18 and CHX19 in yeast, and expression in E coli.

To date, this is the only study showing AtCHX17 can complement yeast mutant *kha1p* functionally.

Please cite:

Plant-specific cation/H⁺ exchanger 17 and its homologs are endomembrane K⁺ transporters with roles in protein sorting.

Chanroj S, Lu Y, Padmanaban S, Nanatani K, Uozumi N, Rao R, Sze H.

J Biol Chem. 2011 Sep 30;286(39):33931-41. doi: 10.1074/jbc.M111.252650.

Reply

The above citation was added to the supplementary information (see Supplementary Information p. 21, citation 24).

Manuscript section: Typo in Fig. 2

Fig. 2. CPA2 tree. Label at right side: is it 'GerN', instead of GreN?

Reply

Typo was corrected.

Overall, this ms makes an important contribution as EcNhaA has been the leading model of like transporters which are found in all domains of life. Yet the mode of transport of CPAs and their biological roles in eukaryotes are largely unknown. The paper highlights key residues and predicts transport mode of most sequenced CPAs on earth. Moreover, experiments showed an electrogenic antiporter can be changed to an electroneutral one via mutation of a few strategic residues.

Reply

Thank you for the kind summary.